# Antarctic Ice Sheet grounding line discharge from 1996 through 2023

Benjamin J. Davison[1], Anna E. Hogg[1], Thomas Slater[2], Richard Rigby[1], Nicolaj Hansen[3]

[1]School of Earth and Environment, University of Leeds, Leeds, LS2 9JT, UK
[2]Department of Geography and Environmental Sciences, Centre for Polar Observation and Modelling, Northumbria University, Newcastle-Upon-Tyne, NE1 8ST, UK.
[3]National Center for Climate Research, Danish Meteorological Institute, Sankt Kjelds Plads 11, Copenhagen Ø, DK-2100, Denmark.

*Correspondence to*: Benjamin J. Davison (b.davison@leeds.ac.uk)

**Abstract.** Grounding line discharge is a key component of the mass balance of the Antarctic Ice Sheet. Here we present an estimate of Antarctic Ice Sheet grounding line discharge from 1996 through to January 2024. We calculate ice flux through 16 algorithmically-generated flux gates across 998 ice sheet, glacier, ice stream and ice shelf drainage basins. We draw on a range of ice velocity and thickness data to estimate grounding line discharge. For ice thickness, we use four bed topography datasets, two firn models and a time-varying ice surface. For the ice velocity, we utilise a range of publicly-available ice velocity maps at resolutions ranging from 240 x 240 m to 1000 x 1000 m, as well as new, 100 x 100 m monthly velocity mosaics derived from intensity-tracking of Sentinel-1 image pairs, available since October 2014. Our dataset also includes the contributions to discharge from changes in ice thickness due to surface lowering, time-varying firn air content and surface mass change between the flux gates and grounding line. We find that Antarctic Ice Sheet grounding line discharge increased from 2,140 ± 189 Gt yr$^{-1}$ to 2,283 ± 207 Gt yr$^{-1}$ between 1996 and 2024, much of which was due to acceleration of ice streams in West Antarctica but with substantial contributions from ice streams in East Antarctica and glaciers on the Antarctic Peninsula. The errors in our discharge dataset stem approximately equally from errors in the underlying ice velocity and thickness measurements; however, there is a large spread in discharge estimates depending on the choice of bed topography dataset and flux gate location. Together, these uncertainties account for much of difference between our results and previous studies. In many places, especially in parts of East Antarctica, substantial (>50 % in some basins) modifications to the ice thickness are required in order to reproduce independent estimates of mass change: this highlights the difficulties remaining in reconciling independent estimates of mass change at the basin-scale, owing to unknown uncertainties in each component of the mass change estimates. It is our intention to update this discharge dataset each month, subject to continued Sentinel-1 acquisitions and funding availability. The dataset is freely available at https://zenodo.org/doi/10.5281/zenodo.10051893 (this manuscript was prepared using version 5 of the dataset) (Davison et al., 2024).

## 1 Introduction

The rate of mass loss from the Antarctic Ice Sheet has accelerated since the early-1990s (Otosaka et al., 2023; Diener et al., 2021; Slater et al., 2021; Shepherd et al., 2019). Mass loss has been greatest and most rapid in West Antarctica, where ice

streams draining into the Amundsen Sea Embayment have accelerated dramatically during the satellite era (Mouginot et al., 2014; Konrad et al., 2017). As such, the majority of mass loss from the Antarctic Ice Sheet is attributed to increases in grounding line discharge – the flux of ice into ice shelves or directly into the Southern Ocean from the grounded Antarctic Ice Sheet (henceforth 'discharge'). Grounding line discharge is therefore a key component for quantifying the 'health' of the Antarctic Ice Sheet, particularly when combined with surface mass balance (SMB) estimates to determine overall ice sheet mass change (Rignot et al., 2019; Sutterley et al., 2014). This 'mass budget' or 'input-output' approach to measuring ice sheet mass change compliments other ice sheet mass change measurements derived from altimetry measurements (Smith et al., 2020; Shepherd et al., 2019) or gravimetric approaches (Diener et al., 2021; Velicogna et al., 2020; Sutterley et al., 2020). The principle benefits of the input-output method are two-fold. Firstly, it permits direct partitioning of mass change between SMB and discharge, which provides insight into the processes driving ice sheet mass change. Secondly, discharge is derived from ice velocity and thickness datasets, which can now be generated through continuous satellite-based monitoring at relatively frequent (~monthly) intervals at the continent scale. These data are available at higher spatial resolution than the other mass change measurement approaches, making the input-output method particularly useful in smaller drainage basins and in mountainous terrain, where it is limited only by SMB model performance. Despite their utility, grounding line discharge measurements for Antarctica are relatively sparse (Rignot et al., 2019; Gardner et al., 2018; Miles et al., 2022; Depoorter et al., 2013) resulting in only one estimate of ice sheet mass change using the input-output method (Rignot et al., 2019; Otosaka et al., 2023; Shepherd et al., 2018), which means that independent verification of ice sheet mass balance using this method is lacking. Furthermore, the limited available discharge estimates feeding into those mass change calculations disagree in some regions and basins (for example, the Antarctic Peninsula) such that opposing conclusions regarding basin-scale mass change must be reached for those basins (Hansen et al., 2021).

Here, we present a new grounding line discharge dataset for the Antarctic Ice Sheet. We draw on several bed topography products and velocity measurements from 1996 through to January 2024, and we use time-varying rates of ice surface elevation change and firn air content. The velocity measurements range in spatial resolution from 1 x 1 km annually to 100 x 100 m every month since October 2014, thereby increasing the detail and frequency of continent-wide discharge estimates over time. We provide these discharge estimates integrated over every published basin definition available for Antarctica – ranging in scale from the whole ice sheet down to 1 km-wide glaciers on the Antarctic Peninsula. It is our intention to update this discharge dataset each month, subject to continued Sentinel-1 acquisitions and funding availability. In addition, we will endeavour to provide irregular updates following the release of new bed topography datasets, grounding lines and if any bugs are identified.

## 2 Data and Methods

### 2.1 Bed topography, ice surface and ice thickness

We estimate grounding line discharge using multiple bed elevation datasets. Our primary estimates of bed elevation and bed elevation error draw predominantly on BedMachine v3 (Morlighem, 2020; Morlighem et al., 2020), but we replace the

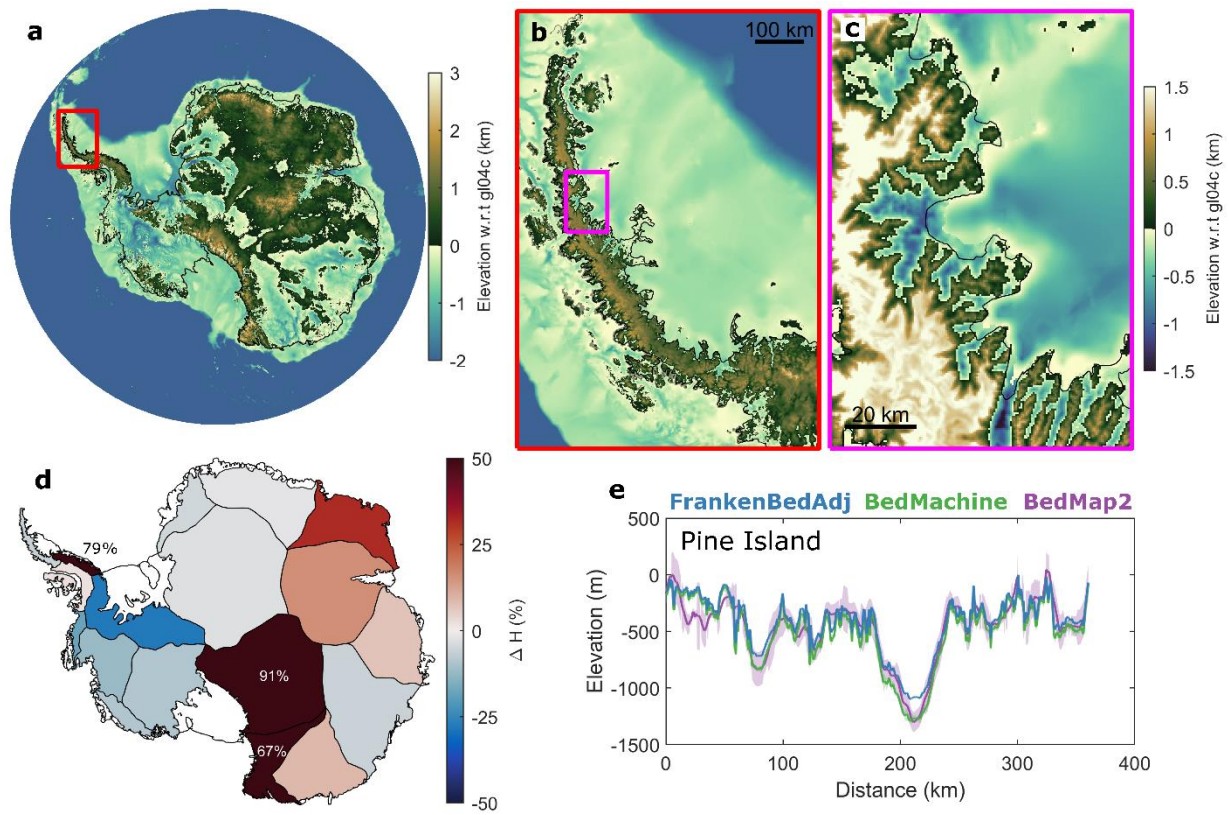

**Figure 1. Antarctic Ice Sheet bed topography overview. (a)** Overview of BedMachine v3. Also shown are overviews of **(b)** the Antarctic Peninsula (Huss and Farinotti, 2014) and **(c)** the Larsen-B Embayment with FrankenBed. **(d)** The change in ice thickness in each MEaSUREs regional basin in FrankenBedAdj compared to FrankenBed, where positive values indicate an increase in ice thickness. **(e)** A comparison of bed elevation in BedMachine, FrankenBedAdj and BedMap2 for Pine Island Glacier at our most seaward flux gate. The coastline and grounding line in panels (a) to (d) are also shown as black lines.

BedMachine bed and bed error with a dedicated bed topography dataset over the Antarctic Peninsula (Huss and Farinotti, 2014), after conversion to a common geoid (GL04c). We use the MATLAB tool wgs2gl04c to perform this conversion (Greene et al., 2019). Henceforth, we refer to this merged bed topography dataset as 'FrankenBed' (Fig. 1; Figs. A2 & A3). We also provide discharge estimates using the bed topography data and associated error from an unmodified version of BedMachine v3 and using BedMap2 (Fretwell et al., 2013).

For each of these bed products, we calculate ice thickness using the Reference Elevation Model of Antarctica (REMA) Digital Elevation Model (DEM), posted at 100 x 100 m and timestamped to 9[th] May 2015 (Howat et al., 2019). Before calculating ice thickness, we reference the REMA DEM elevations to the GL04c geoid and remove the climatological mean (1979-2008) firn air content (Veldhuijsen, Sanne et al., 2022) (Section 2.4). Henceforth, we refer to this firn-corrected ice surface as our reference ice surface, which we assume has a spatially uniform 1 m error (Howat et al., 2019). For the thickness grid calculated

using FrankenBed, we fill exterior gaps through extrapolation along ice flowlines using the same method applied to the reference velocity map described in Section 2.3. The purpose of the extrapolation is to ensure that ice thickness estimates are available at each flux gate pixel (Section 2.4). We chose to extrapolate along flowlines rather than using a more conventional nearest-neighbour interpolation because the latter can lead to erroneous or poorly-targeted sampling near shear margins.

Even though we draw on the best available bed topography and ice surface datasets to construct FrankenBed, some ice remains unrealistically thin given the observed ice flow speeds and the resulting discharge is, in places, lower than that implied by the observed rates of surface elevation change and surface mass balance (Figs. A1 & A2). We therefore generate a final bed elevation estimate at each of our flux gates (Section 2.4) at which we adjust the bed elevation such that the average 1996-2021 discharge across each flux gate matches that required to reproduce observed basin-integrated rates of elevation change over the same time period, after accounting for surface mass balance anomalies obtained from three regional climate models. This method is described further in Appendix A. In summary, we use four ice thickness estimates derived from a reference ice surface and four bed elevation datasets (Fig. 1) – BedMap2, BedMachine v3, FrankenBed and FrankenBedAdj.

To generate an ice thickness time-series from each of these baseline thickness estimates, we modify the REMA DEM using observed changes in ice surface elevation from 1992 to 2023 (Fig. A1) derived from satellite radar altimetry following the methods of Shepherd et al. (2019). Because satellite altimetry measurements do not fully observe the ice sheet margins at monthly intervals, we estimate monthly time series of ice surface elevation change by fitting time-dependent quadratic polynomials (Fig. A1) to the  observed surface elevation changes posted on a 5 x 5 km grid at quarterly intervals, which we linearly interpolate to our gate pixels and evaluate at each velocity epoch (Section 2.3). We apply these modelled time-series of elevation change to each reference ice thickness estimate to form time-series of ice thickness at each gate pixel. We quantify the errors in the elevation change by calculating upper and lower bounds to the quadratic fit from the 95 % confidence interval on each of the model coefficients (Section 2.7.3). South of 81.5°, where elevation change measurements are only available since the launch of CryoSat-2 in 2010, we assume static ice thickness rather than extrapolate the historical thinning rates from those observed between 2010 and 2023. Given that the flux gate pixels south of 81.5° only contribute 6 % to the pan-Antarctic discharge and that the applied thickness changes elsewhere around the continent only modify the total discharge by 0.7 %, this choice has little impact on our pan-Antarctic discharge estimate. We then account for temporal variations in firn air content by adjusting the climatological firn air content correction in each flux gate pixel using time-series of firn air content anomalies from two firn models (Section 2.2) at each velocity epoch. For discharge estimates after the last available output from each firn model, we use the monthly firn air content climatology (1979-2008), in order to capture seasonal changes in firn air content. For discharge estimates after January 2023, when our thickness change observations end, we continue to use the quadratic fit. We also assume no changes in bed elevation due to erosion of the substrate or changes in ice thickness due to changes in subglacial melt rates, both of which are expected to be negligible.

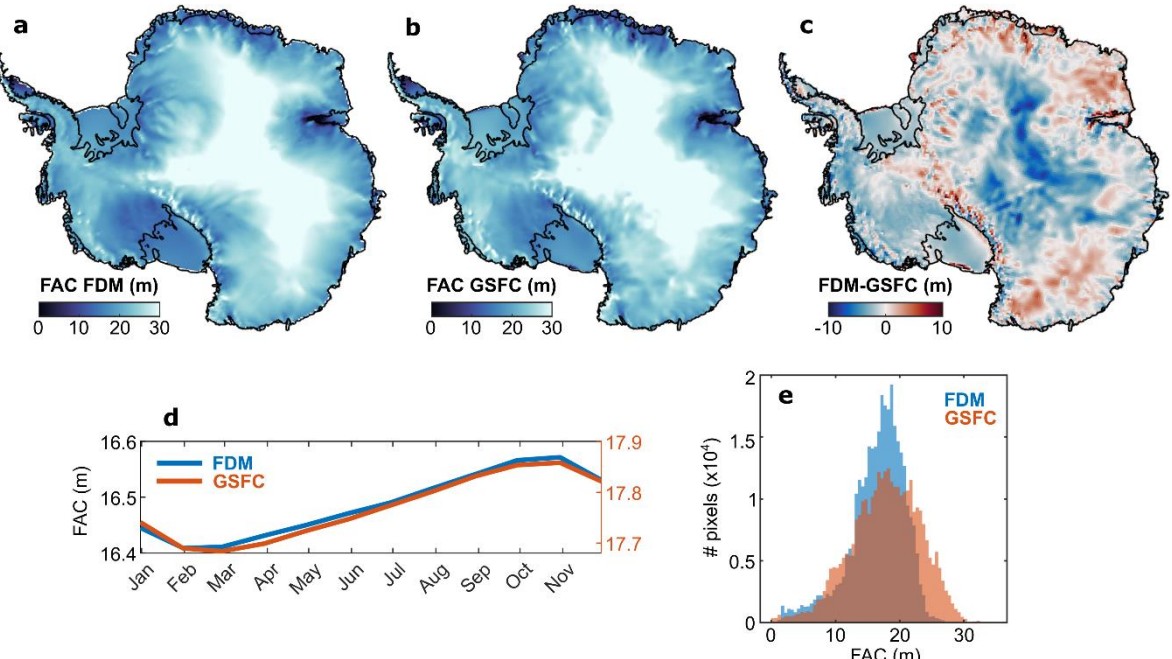

**Figure 2. Overview of firn air content models.** Overviews of (a) the IMAU FDM, (b) the GSFC-FDMv1.2 and (c) the difference between the two models. (d) The climatological seasonal cycle of firn air content (FAC) in each firn model. Note that in panel (d) the IMAU FDM and the GSFC-FDMv1.2 are plotted on separate y-axes to facilitate comparison of their seasonal variability; their units are the same. (e) The frequency distribution of FAC at every flux gate pixel in each model.

## 2.2 Firn air content

We use two firn models (Fig. 2) to remove firn air content from our ice thickness estimates, to determine the ice equivalent thickness at each flux gate and to permit the use of a single ice density value in the discharge calculation (Section 2.7). These are the Institute for Marine and Atmospheric Research Utrecht Firn Densification Model (IMAU FDM) (Veldhuijsen, Sanne et al., 2022) and the Goddard Space Flight Center FDM (GSFC-FDMv1.2), which draws on the Community Firn Model framework and is forced by the Modern-ERA Retrospective analysis for Research and Applications, Version 2 (MERRA-2) climate forcing (Medley et al., 2022b, a). The resolution of the IMAU FDM is 27 x 27 km and the GSFC-FDMv1.2 is 12.5 x 12.5 km. Both models provide daily firn air content for all of Antarctica and span the periods January 1979 to December 2021 for the IMAU FDM and January 1980 to July 2022 for GSFC-FDMv1.2. We use both solutions independently and provide a discharge estimate using each.

## 2.3 Ice velocity

Prior to constructing an ice velocity and discharge time-series, we generate a reference velocity grid in order to fill gaps in the time-series velocity products. We construct the reference velocity grid by combining two velocity products. First, we use a

100 x 100 m multi-year velocity mosaic derived from feature tracking of Sentinel-1 imagery between January 2017 and September 2021 (Davison et al., 2023a). Sentinel-1 imagery are only continuously acquired around the Antarctic Ice Sheet margin, with sparser measurements further inland acquired in 2016. To fill the pole hole in the reference grid, we use the 450 x 450 m MEaSUREs reference velocity product (Rignot et al., 2017), which is linearly interpolated to the grid of the Sentinel-1 product. We fill interior gaps in this mosaic using the regionfill algorithm in MATLAB, which smoothly interpolates inward from the known pixel values on the outer boundary of each empty region by computing the discrete Laplacian over each region and solving the Dirichlet boundary value problem. This interior gap-filling has no bearing on our discharge estimate, but it allows for easier filling of external gaps. We then fill exterior gaps through extrapolation along flowlines following the method of Greene et al. (2022), where the observed velocity is multiplied by the observed thickness mosaic (described in Section 2.1), before extrapolating along the hypothetical direction of flow and inpainting between flowlines. We multiply the ice velocity by the reference ice thickness before extrapolating and inpainting, so as to give appropriate weight to flow directions of thicker ice that contribute more to ice flux. As with the reference thickness map, we choose to extrapolate along flowlines to avoid erroneous sampling of ice velocity, especially near shear margins. This produces a gapless ice velocity map of Antarctica (Fig. 3), broadly representing the average velocity of the ice sheet from 2015 to 2021. We emphasise that the purpose of the gap filling is only to ensure that a velocity estimate is available at every flux gate pixel. As such, the velocity in the ice sheet interior and the extrapolated velocity seaward of the flux gates in this reference map have no bearing on our discharge estimate.

For our time-series product, we compile multiple velocity sources:

1. The 1 x 1 km MEaSUREs annual velocity mosaics (Mouginot et al., 2017b, a) for the year 2000 and from 2005 to 2016

2. Monthly 100 x 100 m velocity mosaics derived from intensity tracking of Sentinel-1 image pairs (described in Appendix B), available from October 2014 to January 2024 (Davison et al., 2023b, a).

3. Monthly 200 x 200 m velocity mosaics derived from intensity and coherence tracking of Sentinel-1 image pairs, available from October 2014 to December 2021 (Nagler et al., 2015).

4. In the Amundsen Sea Embayment in 1996, we also use a combination of 450 x 450 m MEaSUREs InSAR-based velocities derived from 1-day repeat ERS-1 imagery (Rignot et al., 2014), which covers the region spanning Cosgrove to Kohler Glacier, and 200 x 200 m velocities from ERS-1 offset tracking over the Getz basin (https://cryoportal.enveo.at/data/). The latter have been filled using an optimisation procedure supported by the BISICLES ice sheet model (Selley et al., 2021).

5. The 240 x 240 m ITS_LIVE annual mosaics (Gardner et al., 2019) during 1996-2018.

6. Two 450 x 450 m MEaSUREs multi-year velocity mosaics, which incorporate velocity estimates in the periods 1995-2001 and 2007-2009 (Rignot et al., 2022).

7. In the Amundsen Sea Embayment, gap-filled 240 x 240 m ITS_LIVE annual mosaics, from 1996 to 2018 (Paolo et al., 2023; Gardner, 2023)

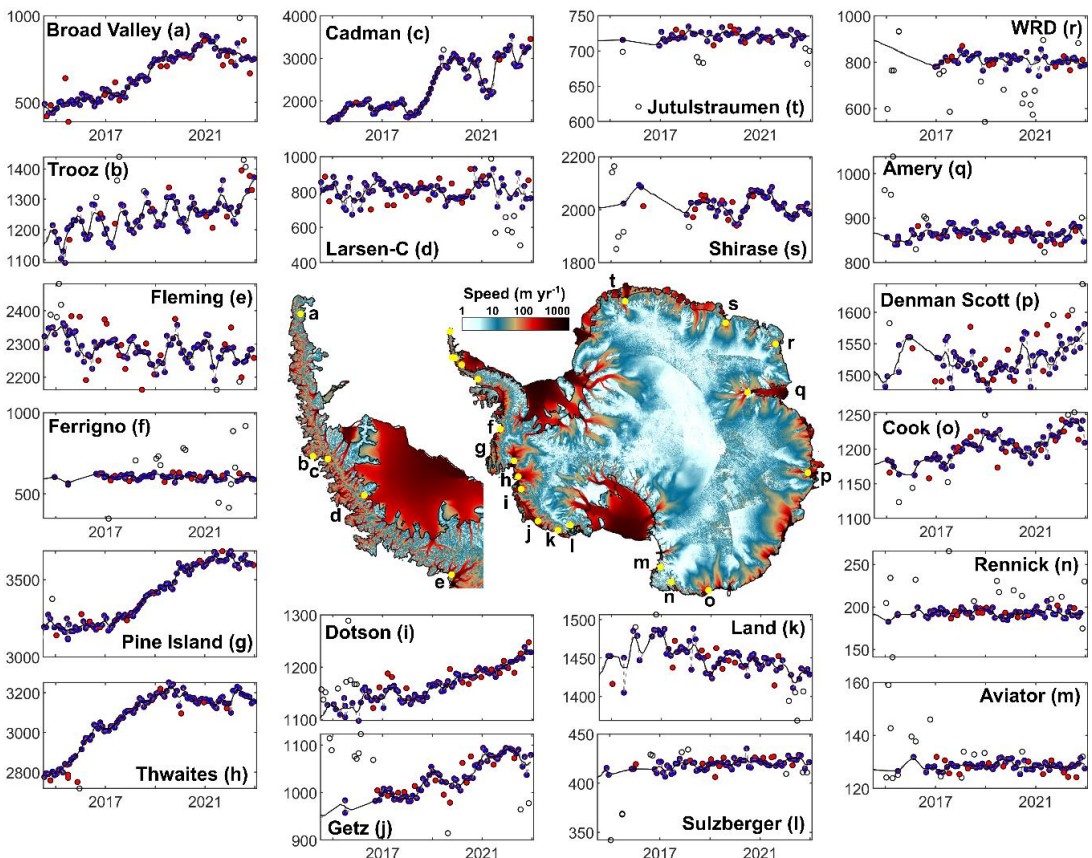

**Figure 3. Reference ice velocity map and time-series outlier removal.** The central plot shows the reference ice velocity map (with extrapolated velocities masked to aid visualisation). Panels (a) to (t) show example time-series of cross-gate velocity (in m yr$^{-1}$) extracted from single flux gate pixels. The black circles show points removed by the global outlier filters, and the red dots show points removed by the local outlier filters are shown. The unsmoothed filled time-series is shown as a grey-dashed line and the smoothed time-series is shown as a black line. WRD is Wilma Robert Downer.

8.  Over Pine Island Glacier, 500 x 500 m mosaics of ice velocity derived from speckle-tracking of TerraSAR-X and TanDEM-X imagery, averaged over 2 to 5 month periods from 2009 to 2015 (Joughin et al., 2021).

Each of these velocity products spans a time period; following Mankoff et al. (2019, 2020), we treat each product as an instantaneous measurement with the timestamp given by the central date in the estimate.

From these data, we generate discharge-ready, gapless velocity time-series at each gate pixel as follows. We linearly interpolate the easting and northing velocities, and their respective errors, from each product to each flux gate pixel. There are consistent differences between velocity data sources that we assume are related to differences in offset tracking parameter choices, digital elevation models used in the tracking, image co-registration procedures, outlier removal routines and final dataset posting. Generally, the datasets posted at a higher resolution, such as the 100 x 100 m Sentinel-1 mosaics and the 240 x 240 ITS_LIVE

mosaics, have higher velocities than the coarser datasets, especially on narrow outlet glaciers. Treating each gate pixel as a time-series, we first remove extreme outliers defined as those more than two-times or less than half of the reference velocity. We then align each data source based on a robust (iteratively re-weighted least-squares) linear fit through their overlapping time-periods, and apply the difference between the means of each fit as a scalar shift to the coarser velocity datasets. This shift increases our pan-Antarctic discharge estimate by 116 Gt yr$^{-1}$ compared to the case where we align the higher resolution datasets down to the coarser datasets.

Treating each flux gate pixel as a time-series, we remove outliers in two stages. Firstly, we remove global time-series outliers after detrending using two passes of a scaled median absolute deviation filter with thresholds of five then three. This global filter is only applied to time-series with more than 30 % of non-nan measurements. Secondly, we remove local outliers using two passes of a moving median filter with a threshold of two median absolute deviations and window sizes of four months then three months.

We fill gaps in each of our flux gate velocity time-series in three stages. Firstly, we linearly interpolate across short temporal gaps (two months or less). Secondly, we linearly interpolate across short spatial gaps (three gate pixels or less). Thirdly, we fill remaining temporal gaps using linear interpolation, then back- and forward-filling at the ends of each time-series. The forward-filling of the velocity time-series is used on all flux gate pixels south of 81.8° during the Sentinel-1 era, which contribute 6.2 % to our Antarctic-wide discharge. For gate pixels with no data at any time and more than three gate pixels from neighbouring finite pixels (after outlier removal), we use our reference ice velocity estimate which has no gaps by definition. This final step affects just 0.05 to 0.15 % of flux gate pixels. After infilling, we smooth each pixel-based time-series with two passes of a moving mean filter, with window sizes of three months then four months. Where we have removed outliers then infilled the time-series, we set the easting and northing error to be |10 %| of the interpolated and smoothed easting and northing velocity components, respectively, at the gate pixel and velocity epoch in question. As in previous studies (Mankoff et al., 2019; Mouginot et al., 2014; Mankoff et al., 2020), we assume the depth-averaged velocity is the same as the measured surface velocity. Examples of this outlier removal and infilling are shown in Fig. 3.

## 2.4 Flux gates

We algorithmically generate 16 flux gates close to the Antarctic Ice Sheet grounding line (Fig. 4). Each flux gate is continuous around the Antarctic Ice Sheet and Wilkins Island; other Antarctic islands are not included in this analysis. The seaward grounding line is placed 3-years of ice flow upstream of the MEaSUREs grounding line (Mouginot et al., 2017c). The ice velocity for this migration is taken from the reference velocity dataset (described in Section 2.3) and the migration is performed in increments of 0.1 years to account for variations in ice velocity along the migration path. Gate pixels are spaced every 100 m for ice flowing faster than 100 m yr$^{-1}$ and 200 m for slower ice, defined on a Polar Stereographic grid (EPSG 3031) and accounting for distance distortions introduced by that projection. 15 additional gates are generated at 200 m increments further

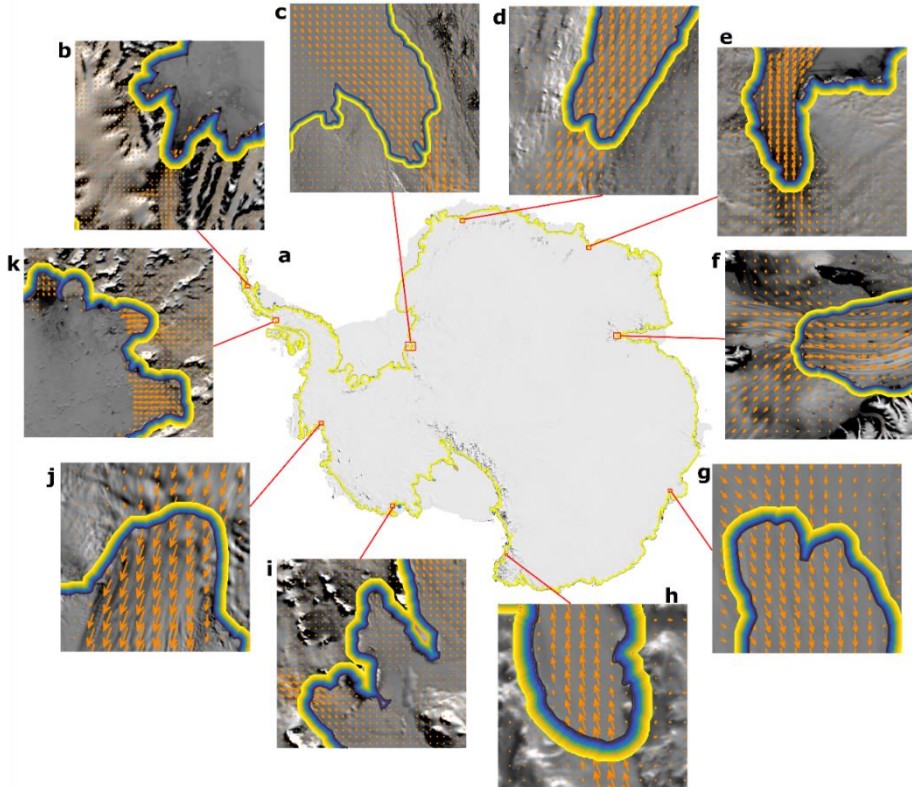

**Figure 4. Flux gate overview.** (a) Overview of Antarctica with flux gates plotted, where yellow lines represent the most inland gate and the blue lines represent the most seaward flux gate. Panels (b) to (k) show zoomed in examples of the 16 flux gates in small regions around Antarctica, with ice velocity vectors overlain (orange arrows). The background image is the MODIS Mosaic of Antarctica (Haran et al., 2018).

upstream of the first gate, such that the most upstream gate is 3 km upstream of the first gate. We provide discharge and error estimates for each of these flux gates and for the mean of all of the gates, weighted by the reciprocal of the error at each gate.

### 2.5 Mass change between flux gates and grounding line

Mass changes occur between each flux gate and the grounding line due to surface processes and due to subglacial melting.
Here, we estimate mass changes due to surface processes only. We estimate this mass change for each drainage basin (Section 2.6) by integrating the climatological (1979-2008) surface mass balance from three regional climate models: RACMO2.3p2 (van Wessem et al., 2018), MAR (Agosta et al., 2019; Kittel et al., 2018) and HIRHAM5 (Hansen et al., 2021) in the area enclosed between each flux gate and the MEaSUREs grounding line (Mouginot et al., 2017c). This mass correction is applied at each velocity epoch. Since surface mass balance is generally positive downstream of the flux gates, this correction increases
our Antarctic-wide grounding line discharge by 64 Gt yr$^{-1}$ on average.

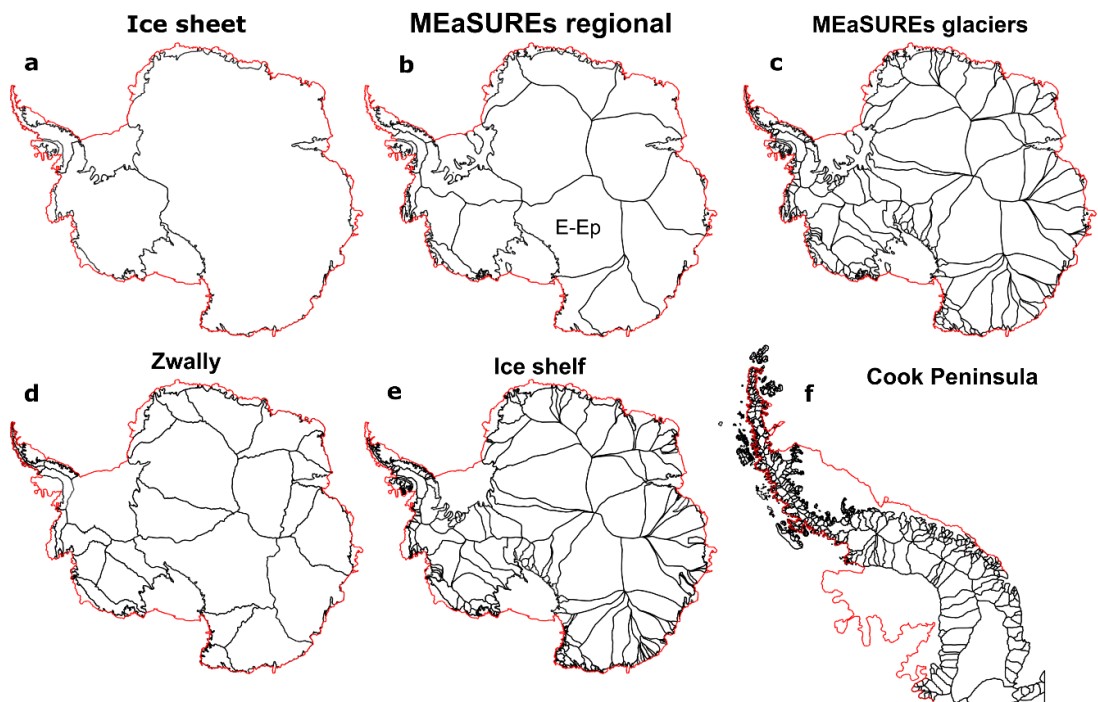

**Figure 5. Overview of Antarctic Ice Sheet drainage basins.** (a) Main ice sheet basins – East Antarctica, West Antarctica and the Antarctic Peninsula. Also shown are smaller drainage basin definitions, including (b) the MEaSUREs regional basins, (c) the MEaSUREs glacier basins, (d) the Zwally basins, (e) ice shelf basins and (f) the Peninsula glacier basins. The coastline is shown in red.

## 2.6 Drainage basins

We provide a discharge estimate for all available Antarctic Ice Sheet basins (Fig. 5). This includes the MEaSUREs regional basins and MEaSUREs glacier basins (Mouginot et al., 2017c), Zwally basins (Zwally et al., 2012), ice shelf basins (Davison et al., 2023a), and Antarctic Peninsula basins (Cook et al., 2014). In total, there are 998 basins used in this study. For each basin, we provide the discharge through each of the 16 flux gates and the average of all flux gates (weighted by the reciprocal of their respective errors) along with their errors. These metrics are provided using each of the four bed topography estimates and with two firn models. In total, therefore, we provide 136 discharge time-series for each basin. In addition, we provide the impact of the two ice thickness corrections – (1) IMAU FDM firn air content and (2) ice surface elevation changes – as well as the impact of downstream surface mass balance on each basin-integrated discharge estimate for each flux gate.

## 2.7 Grounding line discharge

### 2.7.1 Balance discharge

We define the balance discharge as the discharge required to maintain the mass of a given ice sheet basin on 'long' time-scales (decades). In order to maintain the mass of a basin, the hypothetical balance discharge would therefore need to equal the basin-

integrated SMB input on average. Accordingly, we estimate the balance discharge of each basin by integrating the 1979-2008

SMB from the mean of three regional climate models (RACMO2.3p2, MAR and HIRHAM5) within each of the above basins. We estimate the balance discharge error in each basin as the standard deviation of 10 realizations of 20-year climatologies from 1979 to 2008 (i.e. 1979-1999, 1980-2000, etc.). Note that only RACMO2.3p2 is available in 1979.

### 2.7.2 Discharge

We estimate grounding line discharge, $D$, across each flux gate pixel as:

$$D = VHw\rho, \tag{1}$$

where $V$ is the gate-normal ice velocity, $H$ is the ice equivalent thickness, $w$ is the pixel width and $\rho$ is ice density (917 kg m$^{-3}$).

The gate-normal ice velocity is given by:

$$V = sin(\theta)V_x - cos(\theta)V_y, \tag{2}$$

where $V_x$ and $V_y$ are the easting and northing components of the horizontal ice velocity, as defined by the South Polar Stereographic grid (EPSG3031), respectively, and $\theta$ is the angle of the flux gate relative to the same grid. To calculate the total discharge from each basin at each velocity measurement epoch, we simply sum the discharge through each flux gate pixel contained within the basin.

### 2.7.3 Discharge error

The uncertainties in our grounding line discharge stem primarily from errors in the ice velocity and ice thickness estimates. We calculate the cross-gate velocity uncertainty, $V_\sigma$, at each pixel and measurement epoch from the errors in the easting and northing velocity components:

$$V_{xmax} = sin(\theta)(V_x + V_{x\_\sigma}) - cos(\theta)V_y,$$

$$V_{xmin} = sin(\theta)(V_x - V_{x\_\sigma}) - cos(\theta)V_y,$$

$$V_{ymax} = sin(\theta)V_x - cos(\theta)(V_y + V_{y\_\sigma}),$$

$$V_{ymin} = sin(\theta)V_x - cos(\theta)(V_y - V_{y\_\sigma}),$$

$$V_\sigma = \sqrt[2]{(V - V_{xmax})^2 + (V - V_{xmin})^2 + (V - V_{ymax})^2 + (V - V_{ymin})^2}, \tag{3}$$

where $V_{x\_\sigma}$ and $V_{y\_\sigma}$ are the errors in the easting and northing component of the ice velocity, respectively.

The thickness uncertainty, $H_\sigma$, at each measurement epoch and gate pixel is calculated as:

$$H_\sigma = \sqrt[2]{(B_\sigma + 1)^2 + F_\sigma^2 + \Delta H_\sigma^2},$$
(4)

where $B_\sigma$ is the bed elevation error taken from the respective bed elevation products, to which we add 1 meter of ice surface elevation error (Howat et al., 2019). $F_\sigma$ is the error in the firn air content correction, which we assume is 10 % of the correction. $\Delta H_\sigma$ is the error in the applied surface elevation change time-series, which we calculate as:

$$\Delta H_{max} = t(a + a_\sigma)^2 + t(b + b_\sigma) + (c + c_\sigma),$$


$$\Delta H_{min} = t(a - a_\sigma)^2 + t(b - b_\sigma) + (c - c_\sigma),$$

$$\Delta H_\sigma = \left(\frac{\lambda_1}{\lambda_0}\right)\left(\frac{(\Delta H_{max} - \Delta H) + (\Delta H - \Delta H_{min})}{2}\right).$$
(5)

Here, $a$, $b$ and $c$ are the quadratic, linear and intercept coefficients of the quadratic fit to the ice surface elevation change data. $a_\sigma$, $b_\sigma$ and $c_\sigma$ provide the bounds on the 95 % confidence interval for each coefficient. $\lambda_0$ and $\lambda_1$ are the sampling frequency of the fit (monthly) and the original observations (every 140 days) on which the fit is based, which together provide a scaling

factor that prevents the uncertainty in $\Delta H$ scaling with the observational frequency.

Using the uncertainties in ice velocity and ice thickness described above, we calculate the velocity-component of the discharge error, $D_{vel\_\sigma}$, and the thickness-component of the discharge error, $D_{H\_\sigma}$ at each flux gate pixel and each measurement epoch. Both components of the discharge error are calculated in a Monte-Carlo approach with 100 iterations. In each iteration, the time-stamped cross-gate velocity and thickness in each pixel are separately modified using uniformly-distributed random

numbers generated from the time-stamped and pixel-based cross-gate velocity and thickness errors. This produces 200 estimates of grounding line discharge at each measurement epoch and each flux gate pixel: 100 using the range of possible ice velocity values and 100 using the range of possible thickness values. The standard deviation of resulting time-stamped, pixel-based discharge estimates amongst each set of 100 iterations is taken as the velocity- and thickness-components of the discharge error.

The total We define our discharge error, $D_\sigma$, in each flux gate pixel and each measurement epoch, $D_\sigma$, as is calculated as:

$$D_\sigma = \sqrt[2]{D_{vel\_\sigma}^2 + D_{H\_\sigma}^2},$$
(6)

where $D_{vel\_\sigma}$ is the velocity-induced discharge error and $D_{H\_\sigma}$ is the thickness-induced discharge error. Both sources of discharge error are timestamped and calculated at each flux gate pixel. We calculate both the velocity- and thickness-induced discharge errors in a Monte-Carlo approach with 100 iterations. In each iteration, the time-stamped cross-gate velocity and thickness in

each pixel are separately modified using uniformly-distributed random numbers generated from the time-stamped and pixel-based cross-gate velocity and thickness errors. The standard deviation of resulting time-stamped, pixel-based discharge estimates amongst the 100 iterations is taken as the discharge error owing to uncertainties in thickness and cross-gate velocity.

We calculate the basin-integrated discharge error, $D_{basin\_\sigma}$, in two ways. Firstly, we follow Mankoff et al. (2019, 2020) and set the basin-integrated discharge error as the average mean difference between the minimum and maximum possible discharge implied by the thickness and velocity errors described above and the central discharge estimate:

$$D_{basin\_\sigma} = ((D_{max} - D) + (D - D_{min}))/2. \tag{7}$$

These errors are typically 7 to 13 % of the basin-integrated discharge and, because they accumulate the error in every pixel, they represent an upper-bound on the discharge error. Secondly, we also provide the 95 % confidence interval of the gate-mean discharge based on the standard error of the discharge estimates through each of the 16 flux gates. The latter approach provides a measure of the uncertainty in the discharge estimate associated with the gate location, which in turn reflects the errors in the underlying ice velocity and ice thickness datasets, and are typically less than 5 % of the basin-integrated discharge. In the following, all statistics use the former upper bound estimate of discharge error, whilst plots use the latter estimate, to facilitate visualisation of discharge changes.

## 3 Results

### 3.1 Grounding line discharge

We provide grounding line discharge estimates through 16 flux gates using four bed topography products and two firn models for 998 drainage basins. In the following, we primarily present values from the mean of all flux gates (weighted by the reciprocal of their errors) using our favoured bed topography dataset (FrankenBed) and the IMAU FDM. We also present comparisons across gates, bed topography datasets and firn models in turn.

Our primary discharge dataset (Fig. 6) gives a total Antarctic grounding line discharge of 2,140 ± 189 Gt yr$^{-1}$ in July 1996, rising to 2,283 ± 207 Gt yr$^{-1}$ in January 2024. On average, Antarctic discharge has increased at a rate of 4.9 Gt yr$^{-2}$ or 0.2 % yr$^{-1}$ over the study period from 1996. Our dataset shows that Antarctic grounding line discharge has not risen steadily during our study period. Discharge increased steadily from 1998 to 2012 and since 2018. These periods of rising discharge were interrupted by a period of steady discharge.

Our dataset also provides grounding line discharge measurements for distinct Antarctic regions (Fig. 6). Grounding line discharge from West Antarctica increased from 841 ± 72 Gt yr$^{-1}$ in July 1996 to 946 ± 81 Gt yr$^{-1}$ in January 2024, with a trend of 3.8 Gt yr$^{-2}$ or 0.4 % yr$^{-1}$ and following a similar pattern of temporal variability described above. West Antarctica therefore currently accounts for approximately 41 % of all Antarctic grounding line discharge and 73 % of the total Antarctic increase in discharge from 1996 through 2023. Discharge from East Antarctica also increased, from 1,000 ± 87 Gt yr$^{-1}$ in 1996 to 1,021 ± 94 Gt yr$^{-1}$ in January 2024, with a statistically significant trend of 0.55 Gt yr$^{-2}$. However, East Antarctic discharge is the most uncertain of any region and fluctuated on approximately 10-year time-scales with an amplitude of approximately 20 Gt yr$^{-1}$. This relative large uncertainty and temporal variability means that East Antarctic grounding line discharge during 2011 to

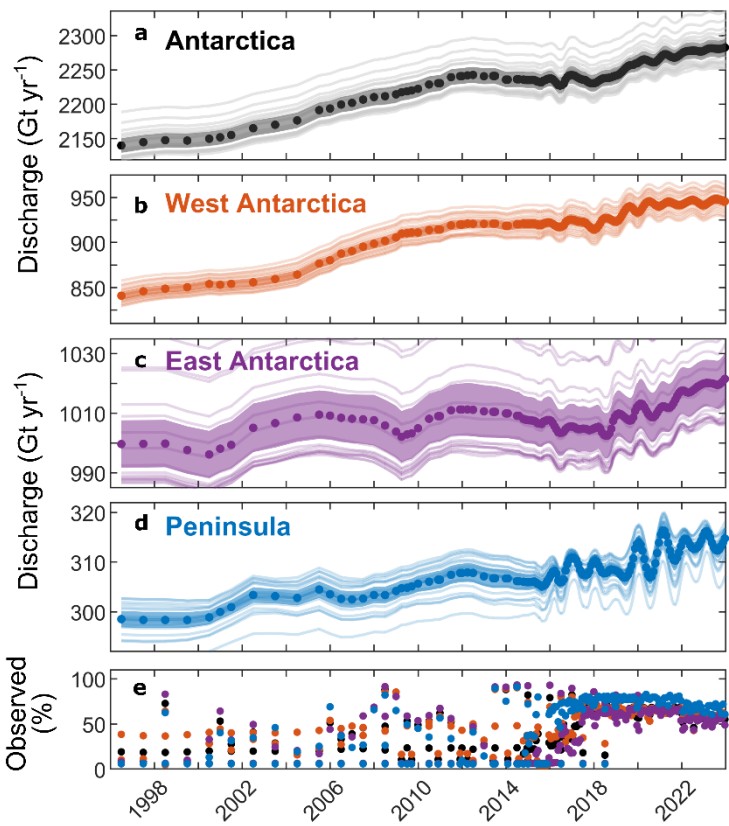

**Figure 6. Antarctic Ice Sheet grounding line discharge.** Discharge time-series for (a) Antarctica, (b) West Antarctica, (c) East Antarctica and (d) the Antarctic Peninsula. In each panel, the dots show the central discharge estimate with 95 % confidence bounds (shading) and the discharge through each individual flux gate (faint lines). (e) The proportion of discharge that is observed, as opposed to infilled, shown for the whole Antarctic Ice Sheet (black dots), West Antarctica (orange dots), East Antarctica (purple dots) and the Antarctic Peninsula (blue dots).

2015 was not significantly different from that during 2002 to 2008, and may explain previous reports of unchanging East Antarctic grounding line discharge that were based on comparisons between two epochs during those periods (Gardner et al., 2018). Grounding line discharge from the Antarctica Peninsula was $299 \pm 29$ Gt yr$^{-1}$ in 1996, increasing to $313 \pm 32$ Gt yr$^{-1}$ on average during April to September 2023, with a significant trend of 0.6 Gt yr$^{-2}$ or 0.2 % yr$^{-1}$. Our monthly discharge estimates since 2015 contain pronounced seasonal variations in discharge on the Antarctic Peninsula as a whole and on many of its outlet glaciers, as shown by two other studies to date (Boxall et al., 2022; Wallis et al., 2023). The seasonal cycles across the whole Peninsula have an amplitude of approximately 5-10 Gt yr$^{-1}$ but with substantial variability between years (Fig. 6).

Within the above regions, we provide discharge time-series for individual glacier, ice stream and ice shelf basins. A selection of these basins, spanning discharges from less than 0.1 Gt yr$^{-1}$ to over 100 Gt yr$^{-1}$, are shown in Fig. 7. The top five contributors to Antarctic-wide grounding line discharge, on average since 2016, are Pine Island Glacier ($146 \pm 4$ Gt yr$^{-1}$), Thwaites Glacier

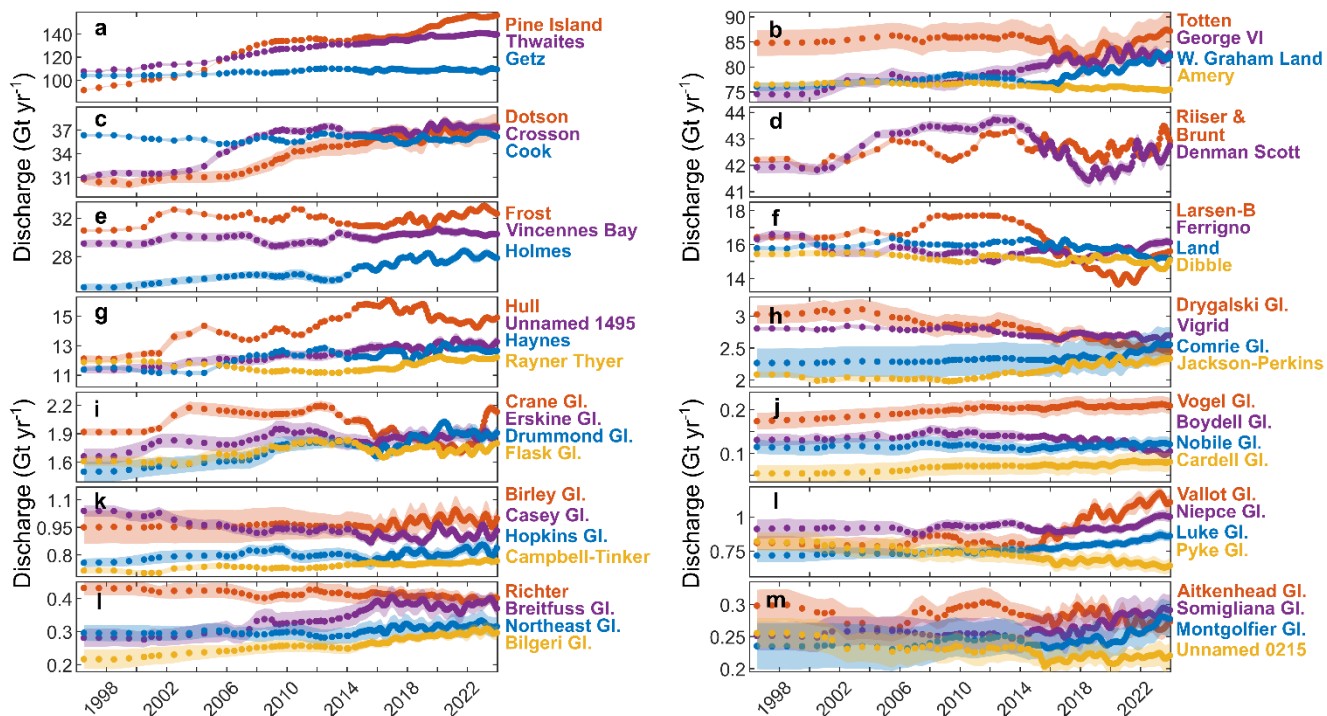

**Figure 7. Basin-scale grounding line discharge examples.** Grounding line discharge for selected basins from 1996 through 2023. The points show the gate-average discharge estimate and the shading shows the discharge uncertainty (95 % confidence limits). Glacier locations in Figure 8.

(137 ± 2 Gt yr⁻¹), Getz drainage basin (109 ± 2 Gt yr⁻¹), Totten Glacier (84 ± 0.3 Gt yr⁻¹) and George VI (77 ± 1 Gt yr⁻¹). Discharge from Pine Island Glacier increased from 91 ± 7 Gt yr⁻¹ to 156 ± 13 Gt yr⁻¹ from 1996 to January 2024, but this

increase was interrupted by relatively steady discharge from 2009 to 2017 and since 2022 (Fig. 7a). Our dataset also includes other well-known changes in grounding line discharge around Antarctica, including increases at Thwaites Glacier, Crosson and Dotson ice shelves (Fig. 7a,b), and a progressive deceleration of the Larsen-B tributary glaciers until their recent acceleration in 2022 (Fig. 7f; Ochwat et al., 2023; Surawy-Stepney et al., 2023) and Whillans Ice Stream (Fig. 8; Joughin et al., 2005). Our dataset also reveals substantial changes in discharge at many glaciers and ice shelves that are less well-known.

These include, for example, increases in discharge from Cook Ice Shelf basin (Miles et al., 2022), Muller Ice Shelf, Denman Scott Glacier, Holmes, Vincennes Bay (primarily from Vanderford Glacier), Frost Ice Shelf, Ferrigno Ice Shelf, as well as numerous glaciers on the Antarctic Peninsula (Fig. 7). Other basins, such as Richter, Dibble and Boydell Glacier, show declining discharge, whilst many others, such as Rayner Thyer, undergo large multi-year fluctuations in discharge (Fig. 7).

Fig. 8 provides an overview of 1996 to 2024 trends in grounding line discharge from individual glacier and ice stream basins

around Antarctica. This overview highlights the rapid increase in grounding line discharge from the Amundsen Sea

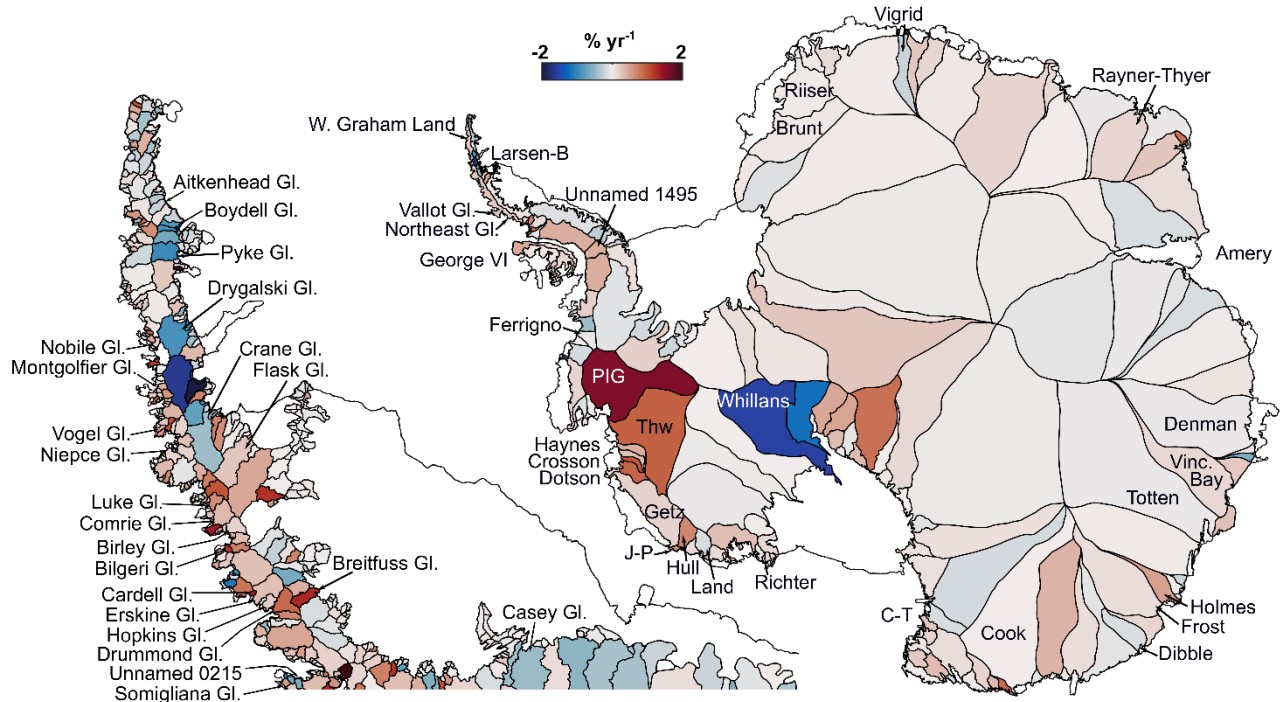

**Figure 8. Basin-scale grounding line discharge trends from 1996 to 2023.** Overview of grounding line discharge trends from 1996 through 2023, as a percentage of the median discharge in each drainage basin. Basins mentioned in the main text and Figure 7 are labelled. Some basin names have been shortened for display purposes: "Vinc. Bay" is Vincennes Bay; "PIG" is Pine Island Glacier; "Thw" is Thwaites; "Riiser" is Riiser-Larsen; "C-T" is Campbell-Tinker; and "J-P" is Jackson-Perkins.

Embayment of West Antarctica, as well as weaker increases in the Bellingshausen Sea, the west coast of the Antarctica Peninsula and across the Indian Ocean-facing sector of Antarctica. It also shows declines in grounding line discharge from Whillans Ice Stream, from numerous basins around Amery Ice Shelf in East Antarctica, and from many glaciers on the east coast of the Antarctic Peninsula (Fig. 8). This broad spatial pattern of grounding line discharge change is consistent with, but

adds more detail to, changes in ice sheet surface elevation over a similar time period (Shepherd et al., 2019).

### 3.2 Effect of bed topography dataset on discharge

Excluding FrankenBedAdj, the choice of bed topography dataset affects the Antarctic-wide discharge estimate by 55 Gt yr$^{-1}$ on average (Fig. 9). At the continent scale, FrankenBed produces the highest discharge and BedMachine and BedMap2 respectively produce discharge 3.0 % and 4.2 % lower than with FrankenBed. BedMap2 gives discharge that is 3.5 % lower

than FrankenBed in West Antarctica, 1.6 % greater in East Antarctica and 26 % lower on the Antarctic Peninsula. BedMachine and FrankenBed are identical in West Antarctica and East Antarctica, but BedMachine gives discharge 22 % lower than FrankenBed on the Peninsula. Within individual MEaSUREs glacier basins, BedMachine typically causes either positive or negative discharge changes of 2.5 % or less compared to FrankenBed and the discharge implied by BedMap2 is typically lower

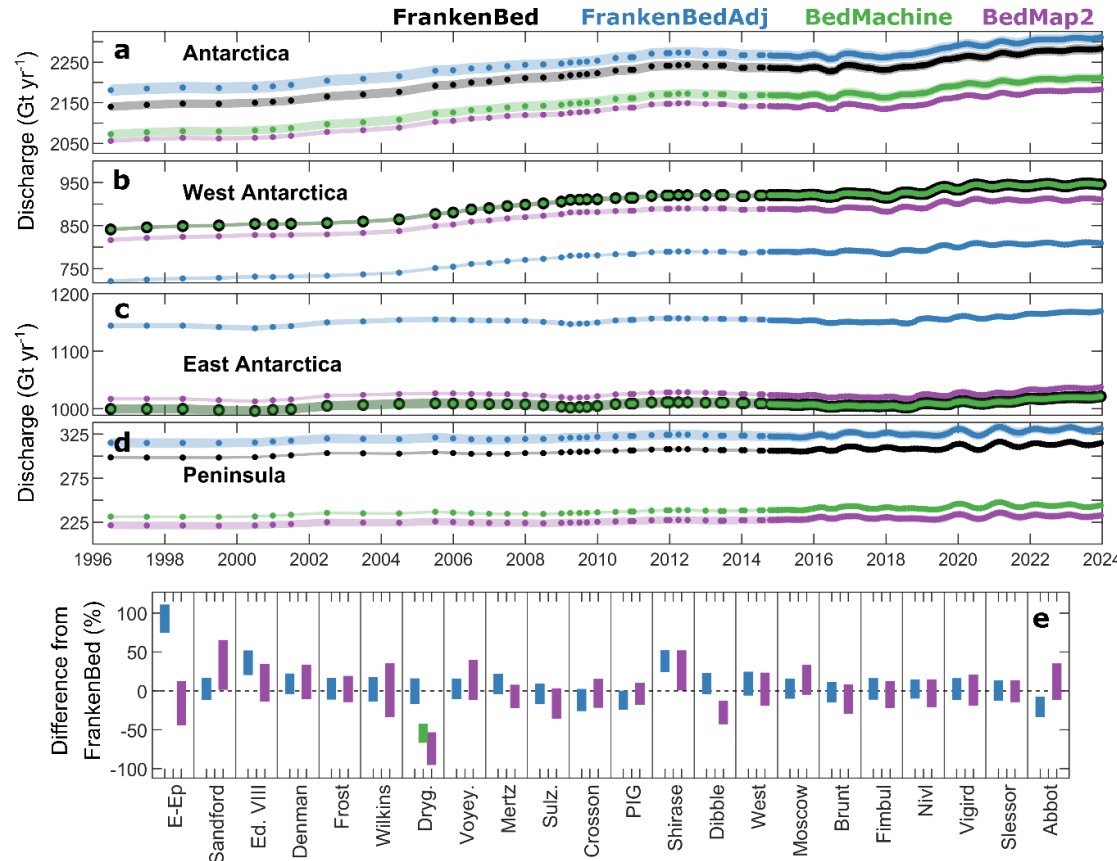

**Figure 9. Impact of bed topography dataset on grounding line discharge.** Grounding line discharge time-series averaged across all flux gates for (a) Antarctica, (b) West Antarctica, (c) East Antarctica and (d) the Antarctic Peninsula. Panel (e) shows the percentage difference in grounding line discharge produced using FrankenBedAdj, BedMachine and BedMap2 compared to FrankenBed, for a range of drainage basins. The vertical extent of each bar represents the potential spread in the differences between bed products owing to error in each discharge estimate. Note that FrankenBed and BedMachine are identical for all displayed basins except Drygalski Glacier. Some basins have been shortened for display purposes: "Ed. VIII" is Edward VIII, "Dryg." is Drygalski Glacier, "Voyey." is Voyeykov Ice Shelf and "Sulz." is Sulzberger Ice Shelf.

than from FrankenBed by discharge by 1 to 5 % (Fig. 9). The impact can be much larger for some individual basins, especially
those on the Antarctic Peninsula; for example, the discharge from Drygalski Glacier is over 40 % lower using BedMachine
and BedMap2 than it is with FrankenBed. The standard error of discharge across our 16 flux gates is similar between
FrankenBed, BedMachine and BedMap2, despite the increase in bed topographic observations and improvements in
interpolation and assimilation methods since BedMap2 was developed.

Our grounding line discharge estimate derived using FrankenBedAdj differs substantially from that using the other bed
products in the majority of basins (Fig. 9). FrankenBedAdj increases our pan-Antarctic discharge estimate by 1.4 %, but has

opposing and roughly equal effects in East and West Antarctica; decreasing West Antarctic discharge by 14 % whilst increasing East Antarctic discharge by 14.5 %. It increases discharge from the Peninsula by 5 %. For some basins, the impact of FrankenBedAdj is dramatic, for example discharge from basin E-Ep in East Antarctica (location in Fig. 5), is over 80 % greater using FrankenBedAdj than that with FrankenBed (Fig. 9). Some of the differences between FrankenBedAdj and the other bed
topography products will be due to uncertainties in mass balance estimates and unknown uncertainties in SMB modelling, particularly in and around basin E-Ep, and in mountainous areas like the Antarctic Peninsula, where radar-derived elevation change measurements have lower performance and SMB models disagree substantially (Mottram et al., 2021). Nevertheless, basins in which the discharge from FrankenBedAdj differ substantially from FrankenBed (Fig. A2) could be useful areas to target future bed topographic mapping campaigns. We reiterate that the derivation of FrankenBedAdj assumes that ice
thickness is the only contributor to differences between mass balance estimates derived from the input-output method and altimetry measurements (Appendix A), so we consider these discharge differences upper bounds on that owing to uncertainties in bed topography.

### 3.3 Effect of gate location

Antarctic-wide grounding line discharge varies by 41 Gt yr$^{-1}$ (1.8 %) on average between our most upstream and downstream
flux gates, and individual gates are generally less than 2 % different from the gate-average discharge (Fig. 10). East Antarctic has the largest relative change in discharge between flux gates: discharge from the most seaward gate is 3.7 % greater than the most upstream gate and 2.5 % from the gate-mean discharge. The Antarctic Peninsula exhibits some seasonality in the inter-gate discharge differences (Fig. 10), likely reflecting seasonal changes in velocity retrieval in summer and winter. The differences between flux gates primarily reflects the difficulty in conserving mass with imperfect ice thickness, velocity and
surface mass balance data, rather than algorithmic errors. Reflective of this, the location of the flux gate makes a small difference for basins where the bed is well surveyed. For example, at Pine Island Glacier, the maximum discharge difference between any flux gate and the gate-average is just 2.1 Gt yr$^{-1}$ (1.5 %). Some studies (Gardner et al., 2018; Davison et al., 2023b) have minimised the impact of uncertain bed topography by placing their flux gates directly over bed topographic observations (primarily from radar flight lines). We opt instead to use the inverse error-weighted average of all gates, which
has the advantage of permitting algorithmic gate generation and will prioritise gates positioned closer to bed elevation observations since the error in the bed products is primarily determined by the distance to the nearest bed elevation observation.

### 3.4 Effect of thickness adjustments

We apply two modifications to the reference ice thickness extracted at each flux gate. These are (1) applying observed rates of surface elevation change based on a quadratic fit to elevation change observations from 1992 to 2023 to obtain a time-series
of ice thickness at each flux gate pixel, and; (2) the removal of firn air content using a time-series of firn air content from two firn models. We also correct the basin-integrated discharge to account for surface mass balance changes between each flux gate and the grounding line. Antarctic-wide, the overall impact of these modifications is to increase grounding line discharge

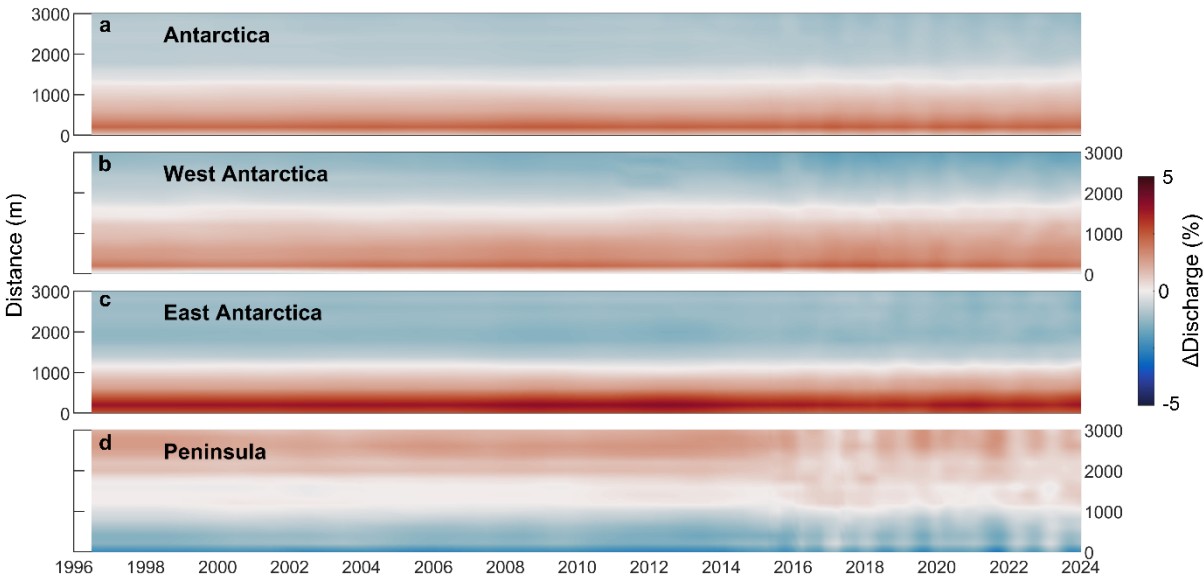

**Figure 10. Impact of flux gate location on grounding line discharge.** Time-series of the percentage difference in grounding line discharge from the inverse error-weighted mean discharge across all flux gates for (a) Antarctica, (b) West Antarctica, (c) East Antarctica and (d) the Antarctic Peninsula. The y-axes correspond to the distance upstream of the first flux gate.

by 23 Gt yr$^{-1}$ in 1996 and reduce it by 10 Gt yr$^{-1}$ in January 2024 (Fig. 11). The individual corrections for firn air content and surface mass balance impacts are larger (over 50 Gt yr$^{-1}$) but opposing and change little over time. The majority of the change
in the impact of these modifications from 1996 through 2024 is due to changes in ice surface elevation during that period, which cause an overall decrease in discharge of 28 Gt yr$^{-1}$ from 1996 through 2023 (Fig. 11). The impact of surface elevation changes on grounding line discharge is greatest in West Antarctica, where thinning rates are highest (Fig. 11; Fig. A1). The impact of firn air content removal is comparable in East and West Antarctica (approximately 22 Gt yr$^{-1}$ or 2 % discharge reductions each, on average) and is greatest in relative terms on the Peninsula (14 Gt yr$^{-1}$ or 4 %). The effect of gate-to-
grounding line SMB changes is to increase Antarctic grounding line discharge by 21 Gt yr$^{-1}$ at the most seaward flux gate, increasing to 105 Gt yr$^{-1}$ at the most upstream gate (Fig. 11).

The choice of firn densification model has a negligible (0.4 %) impact on Antarctic-wide grounding line discharge (Fig. 12), regardless of which flux gate is used. The IMAU-FDM gives consistent lower firn air content (Fig. 2d) so produces slightly higher discharge values than the GSFC-FDMv1.2. The differences between the firn models are generally greatest (~1 %
discharge equivalent) on the Peninsula, which we interpret to be primarily due the ability of each model to resolve the impact of steep topography on surface processes, owing to their different spatial resolutions (12.5 x 12.5 km for the GSFC-FDMv1.2 and 27 x 27 km for the IMAU FDM). In some basins, the choice of firn model makes an appreciable difference – for example, at Moser Glacier, the IMAU-FDM decreases grounding line discharge by 4 % relative to the GSFC-FDMv1.2 on average.

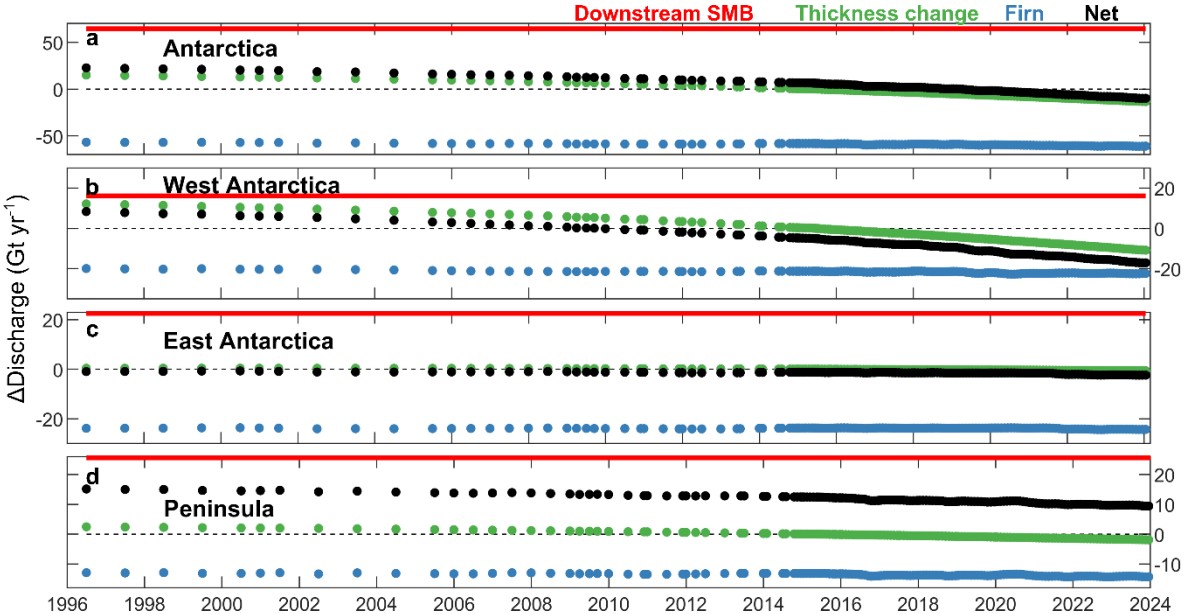

**Figure 11. Timeseries of ice thickness and surface mass balance (SMB) corrections.** The impact of SMB changes downstream of the flux gate (red dots), altimetry-derived thickness change (green dots) and removal of firn air content (blue dots) (described in text) on the derived grounding line discharge from (a) Antarctica, (b) West Antarctica, (c) East Antarctica and (d) the Antarctic Peninsula. The sum of the three corrections is also shown (black dots). Note that surface elevation changes are applied to our reference Antarctic Ice Sheet surface, which is timestamped to 9th May 2015.

Basins with large relative differences are generally very small – with widths much less than the resolution of either firn model – so contribute little to total Antarctic discharge and require extraction from a single firn model pixel that will in many cases not resolve the glacier geometry. Overall then, the use of a firn model has a large enough impact on grounding line discharge to be relevant to glacier mass balance, but the choice of firn model seems to have little impact on Antarctic discharge, at least for the two firn models examined here.

## 4 Discussion

### 4.1 Comparison to previous estimates

Surprisingly few estimates of Antarctic grounding line discharge have been published and made freely available, so we hope that the community will benefit from the release of the dataset described in this study. We focus our comparison on previous estimates that encompass the majority or all of the Antarctic Ice Sheet (Gardner et al., 2018; Rignot et al., 2019; Depoorter et al., 2013; Miles et al., 2022). We note that the '2008' discharge estimates from Gardner et al. (2018) and Depoorter et al. (2013) were estimated using a velocity mosaic (Rignot et al., 2017) compiled from images acquired during the 1996 to 2009 period, but the majority of those images were acquired between 2007 and 2009. To compare our discharge time-series to those

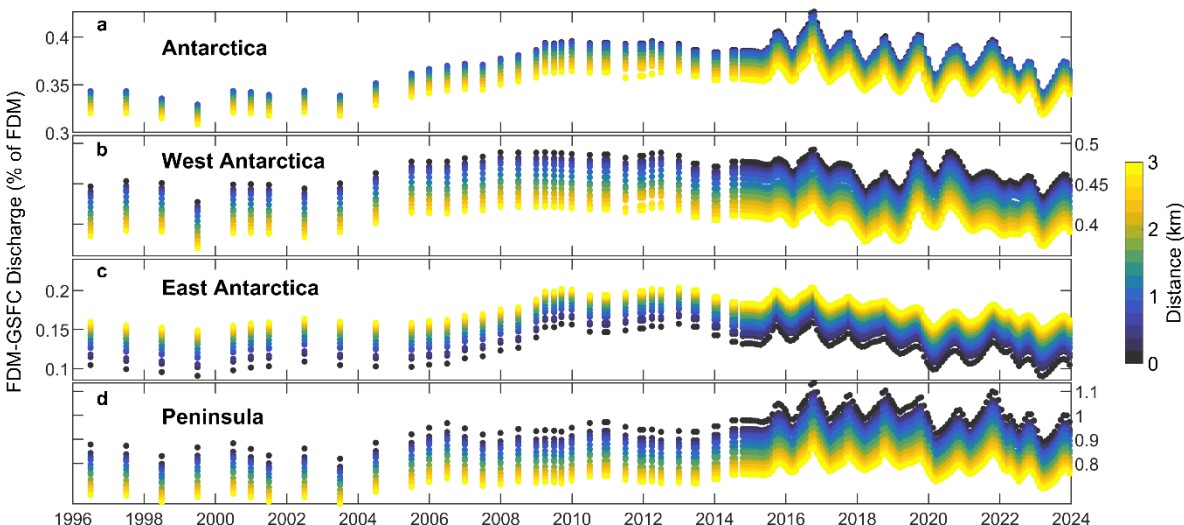

**Figure 12. Impact of firn model choice on Antarctic grounding line discharge.** Time-series of the difference in grounding line discharge when using the IMAU FDM compared to the GSFC-FDMv1.2 from (a) Antarctica, (b) West Antarctica, (c) East Antarctica and (d) the Antarctica Peninsula. Point are coloured according to their distance in kilometres from the most downstream flux gate (blue) compared to the most inland flux gate (yellow).

data, we use our average discharge from January 2007 to December 2009. Rignot et al. (2019) used a range of methods to estimate grounding line discharge; we restrict our comparison to basins for which discharge was estimated using a comparable method (i.e. using both measured ice velocity and ice thickness).

There are substantial differences between our discharge estimates and other published estimates for some basins and for Antarctica, East Antarctica, West Antarctica and the Antarctic Peninsula as a whole (Fig. 13). For Antarctica as a whole, our new dataset compares favourably to that of Rignot et al. (2019) but our dataset gives discharge on average 133 Gt yr$^{-1}$ greater in West Antarctica, 97 Gt yr$^{-1}$ lower in East Antarctica and 13 Gt yr$^{-1}$ lower on the Antarctic Peninsula than estimates from Rignot et al. (2019). In some basins, the differences among estimates are large relative to our estimates dicharge. For example,

Depoorter et al. (2013) estimate the discharge from Filchner-Ronne to be 72 Gt yr$^{-1}$ (24 %) lower than in this study, the discharge from Brunt and Riiser-Larsen to be 5.6 Gt (13 %) greater, the discharge from Pine Island to be 24 Gt yr$^{-1}$ (18 %) lower, and the discharge from Sulzberger to be 4 Gt (24 %) greater than in this study. Depoorter et al. (2013) use a combination of ice thickness estimates based on the assumption of hydrostatic equilibrium and from ice penetrating radar measurements, whereas we draw on gridded bed topography products – the differences between these thickness datasets accumulate across

long flux gates like that at Filchner-Ronne, Brunt, Riiser-Larsen and Sulzberger. Depoorter et al. (2013) also use different flux gate positions than used here, which could plausibly account for much of the difference in flux estimates. To illustrate, we find a 27 Gt yr$^{-1}$ difference between our most upstream and downstream flux gates at Filchner-Ronne. Some small basins with little grounding line discharge have very large proportional differences between estimates. For example, our discharge into Conger

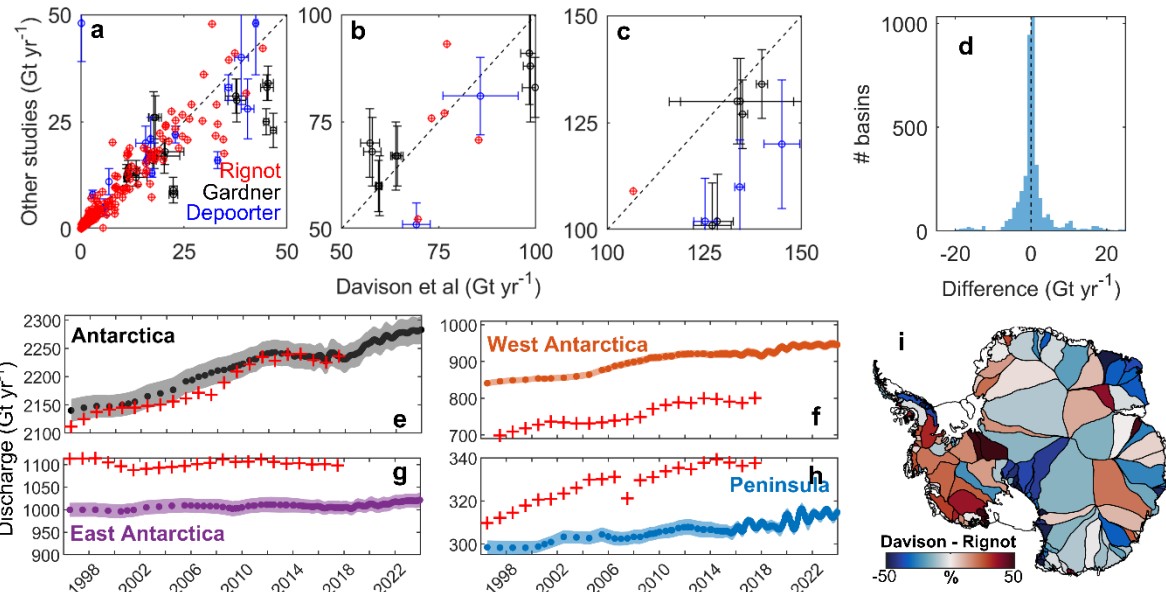

**Figure 13. Comparison to existing grounding line discharge estimates.** Panels (a) to (c) show comparisons with Rignot et al. (2019), Gardner et al. (2018) and Depoorter et al. (2013) for equivalent basins and during overlapping time-periods. Each point shows the discharge for a single basin, with errors provided where available. (d) Histogram of the discharge residuals from panels (a) to (c). Panels (e) to (h) show time-series of our primary discharge estimate (using FrankenBed) compared to estimates from Rignot et al. (2019) (red crosses). (i) Basin-scale comparison of discharge between this study and Rignot et al. (2019) during their overlapping time periods, as a percentage of the Rignot discharge.

Glenzer Ice Shelf is 0.8 Gt yr$^{-1}$, or about 60 % of that in Rignot et al. (2019). In these cases, the absolute differences are comparable to the error in the estimate and may due to the resolution of the flux gate used, as well as the choice of bed topography and ice surface dataset.

## 4.2 Implications for mass budget estimates

At present, only one input-output estimate of Antarctic Ice Sheet mass balance is available (Rignot et al., 2019). This sparsity of input-output data limits the otherwise comprehensive scope of ice sheet mass balance inter-comparison exercises (Otosaka et al., 2023) and limits insights conferred by mass budget partitioning attempts. Here, we briefly examine the mass balance implied by our grounding line discharge and the mean of three regional climate models (Fig. 14), in comparison to a reconciled mass balance estimate (Otosaka et al., 2023). The mass balance of Antarctica and each ice sheet region implied by our discharge estimates using FrankenBedAdj are, perhaps unsurprisingly, very similar to those presented in the latest IMBIE assessment (Fig. 14). FrankenBed compares well to the IMBIE estimate for Antarctica as a whole and on the Peninsula, but overestimates mass loss in West Antarctica and implies large mass gain in East Antarctica, rather than negligible mass change. It is our hope that the community will take advantage this discharge dataset and the full range of available SMB datasets to compute Antarctic

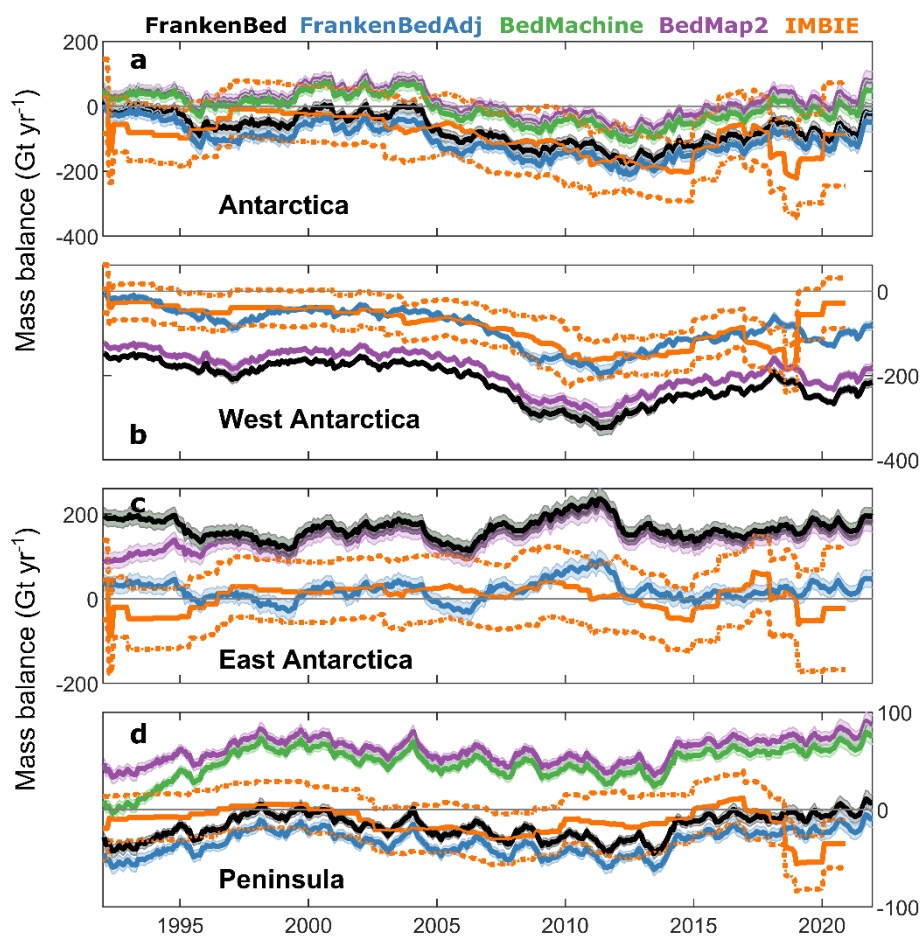

**Figure 14. Antarctic Ice Sheet mass balance.** Mass balance time-series for (a) Antarctica, (b) West Antarctica, (c) East Antarctica and (d) the Antarctic Peninsula, compared to the third IMBIE assessment in orange (Otosaka et al., 2023). Mass balance is calculated using the mean of the three regional climate models described in the main text.

Ice Sheet mass change for any basin of interest, so as to enable deeper investigations into uncertainties and drivers of Antarctic Ice Sheet mass change.

There remains significant uncertainty in mass change estimates of some basins as calculated using the input-output method. In
some basins, the differences between available discharge estimates, including our own, can lead to opposing conclusions regarding the overall mass change of the basin. For example, Hansen et al. (2021) found opposing mass change trends on the Antarctic Peninsula when using two different discharge datasets and a single SMB dataset. This is also apparent in our dataset when comparing the mass balance of the Peninsula computed using different bed topography datasets (Fig. 14). The mass change sensitivity to the choice of discharge dataset will be particularly severe in basins from which the grounding line
discharge is consistently above or below the integrated SMB by an amount comparable to the error in the discharge estimates.

Whilst we think that our grounding line discharge estimate offers many improvements in terms of spatial and temporal resolution, combination of ice velocity datasets, time period surveyed and processes included (firn air content, surface elevation change and surface mass changes), several uncertainties still hinder input-output mass balance estimates. We still find substantial differences in some basins between (i) our primary discharge estimates from FrankenBed compared to those provided by FrankenBedAdj (which we tuned to match independent mass loss observations) and (ii) the discharge estimates provided by Rignot et al. (2019), who manually selected different approaches to determine grounding line discharge. In West Antarctica, our FrankenBed discharge is greater than that in Rignot et al. (2019) and that from FrankenBedAdj. In East Antarctica, the opposite is true: FrankenBed implies lower discharge than in Rignot et al. (2019) and FrankenBedAdj. There are only three plausible explanations for these deviations. (1) Uncertainties in ice thickness cause us to overestimate grounding line discharge in West Antarctica and underestimate it in East Antarctica and on the Peninsula. This is likely to be true on the Peninsula, where the bed is poorly surveyed and the balance thickness used here (Huss and Farinotti, 2014) was calculated using low-resolution SMB estimates that give lower SMB than statistically-downscaled equivalents (Noël et al., 2023). (2) Surface mass balance from the mean of RACMO2.3p2, MAR and HIRHAM5 is too high in East Antarctica and too low in West Antarctica, resulting in apparent input-output mass gain in East Antarctica and mass loss in West Antarctica, even where the grounding line discharge with FrankenBed is correct. This could be the case in West Antarctica and in the Amundsen Sea Embayment, where the bed is well-surveyed. (3) The mass loss estimates from gravimetric and altimetric approaches are too great, possibly because of uncertainties in rates of glacial isostatic adjustment, which both approaches are sensitive to, or uncertainties in the density of snow and ice contributing to the observed changes in surface elevation. This could be the case especially in East Antarctica, where small uncertainties accumulate over large integration areas, and on the Peninsula where altimetry measurements are compromised by heavy snowfall and rapid surface melting as well as steep ice surface slopes. The coarse (~100 km) gravimetric footprint and leakage of mass change signals from the surrounding ocean also introduce large uncertainties, especially over the Antarctic Peninsula (Horwath and Dietrich, 2009; Chen et al., 2015; Hansen et al., 2021).

It is probable that all of these factors contribute to differences between mass budget estimates and that basin-scale comparisons between mass budget approaches will offer deeper insights into their respective uncertainties. If our point (1) - uncertainties in ice thickness - were the sole contributor to differences among mass change estimates, then our thickness adjustments in FrankenBedAdj provide an approximate indication of the magnitude and location of poorly-constrained thickness in existing bed products and could be used to guide future bed topographic mapping efforts. However, given that the magnitude of the thickness adjustments in FrankenBedAdj exceed 50 % in some places, and that these are typically in locations where SMB and firn air content estimates from different regional climate models disagree the most (such as basin E-Ep in East Antarctica), we think that factors (2) and (3) likely also contribute to the differences between mass budget approaches at the spatial scale considered here. This inference is supported by the latest Ice Sheet Mass Balance Intercomparison Exercise (Otosaka et al., 2023), which demonstrated that mass change estimates of East Antarctica from gravimetry and altimetry agree more closely with each other than either do with the single available input-output estimate. In contrast, input-output and gravimetry estimates

of West Antarctic mass change are in relatively close agreement compared to that from altimetry observations. Therefore, it seems likely that all of the points raised above contribute to differences among mass change estimates, but that their effect is location-dependent. This location-dependency likely stems from unknown uncertainties in, for example, bed elevation, local topography, firn density, surface mass balance and glacial isostatic adjustment, and will therefore require smaller-scale mass budget assessments that can account for basin-scale conditions. It is our hope that synergistic use of each mass budget approach, with due consideration for the location-specific uncertainties in each method, will lead to increased confidence regarding the direction and magnitude of mass change around Antarctica, as well as improved understanding of the drivers of that mass change.

## 5 Data availability

The ice sheet basins, balance discharge and grounding line discharge estimates are, for the purposes of review, available at: 10.5281/zenodo.10051893(Davison et al., 2024).

## 6 Conclusions

We present a new grounding line discharge product for Antarctica and all of its drainage basins available from 1996 through to January 2024. The temporal resolution and coverage increases from annual and <25 % respectively in the early years of our dataset to monthly and over 50 % respectively in the latter years of our dataset. We show that grounding line discharge from Antarctica increased from $2,140 \pm 189$ Gt $yr^{-1}$ in 1996, rising to $2,283 \pm 207$ Gt $yr^{-1}$ in January 2024. Much of this grounding line discharge change is due to increasing flow speeds of West Antarctic ice streams, but we also observe large increases in discharge at some basins in East Antarctica, including Holmes, Vanderford Glacier, Denman Scott and Cook Ice Shelf. The high spatial resolution of our ice velocity mosaics since October 2014 allow us to measure substantial seasonal variability and pronounced multi-year trends in discharge even on small ~1 km-wide glaciers draining the Antarctic Peninsula.

Our results broadly agree with other published discharge estimates; however, there are substantial differences in some basins. Broadly, our discharge estimate is greater in West Antarctica and lower in East Antarctica and on the Peninsula than other estimates. These differences generally arise due to uncertainties in bed topography, which can accumulate across long flux gates or which represent a substantial proportion of the discharge across very short flux gates, and due to differences in ice velocity resolution and flux gate position. For some basins, the differences between existing discharge datasets, including our own, is significant enough to have bearing on the mass change of those basins when using the input-output method, particularly in basins which remain close to balance but which are persistently above or below balance. This is particularly acute on the Antarctic Peninsula and in East Antarctica, where deriving estimates of ice thickness, ice velocity, firn air content and surface mass balance are fraught with difficulties owing to the steep topography, narrow glaciers, high snowfall and (in places) intense summertime surface melting.

We find that bed topography remains a potentially large uncertainty in grounding line discharge estimates and therefore has the potential to severely limit the utility of attempts to calculate basin-scale ice mass change using the input-output method. In order to reproduce observed rates of mass loss, we have to modify the bed topography by over 50 % in some basins. This may be realistic in some places where the bed is poorly surveyed; however, uncertainties in SMB and uncertainties in independent estimates of mass change should not be ignored. The progressive increase in ice thickness measurements around Antarctica (Frémand et al., 2023) and the improvements in assimilation and interpolation methods (Morlighem et al., 2020; Ji Leong and Joseph Horgan, 2020) will lead to improved estimates of ice thickness around Antarctica – our workflow is designed to facilitate the addition of new bed topography datasets as they become available and we aim to do so. We have provided this dataset for the scientific community so as to ensure that accurate measurements of Antarctic grounding line discharge remain available routinely for researchers everywhere.

**Appendix A: Making FrankenBedAdj bed topography**

Here we describe the method used to adjust the FrankenBed elevations such that the corresponding discharge estimate produces the observed change in ice surface elevation.

The mass balance, $M$, of an ice sheet or ice sheet basin is given by:

$$M = S - D, \tag{A1}$$

where $S$ is the surface mass balance and $D$ is the grounding line discharge. For each of the MEaSUREs regional basins, we estimate the 1996 to 2021 average mass balance by integrating ice equivalent surface elevation change measurements over each basin (Shepherd et al., 2019). The rates of elevation change from 1992 to 2023 are shown in Fig. A1. Using this rate of mass change, we estimate the average discharge required to produce that mass change (Fig. A2), given the surface mass balance anomalies from three regional climate models including RACMO2.3p2, MAR and HIRHAM5. We refer to this as our altimetry-derived discharge.

For each basin, we proportionally adjust the pixel-based ice thickness based on the difference between our calculated basin-scale discharge and the altimetry-derived discharge. This is akin to rearranging Eq. (4) to:

$$H = D/Vw\rho, \tag{A2}$$

Where $V$ is the 1996 to 2021 average velocity normal to the flux gate in each pixel, D is the altimetry-derived discharge and $w$ is the total length of the gate in each basin. In practice, this is an iterative process because we modify the pixel-based ice thickness to solve for the basin-scale altimetry-derived discharge. The effect of these thickness adjustments are shown in Fig. 2 of the main text and Fig. A2.

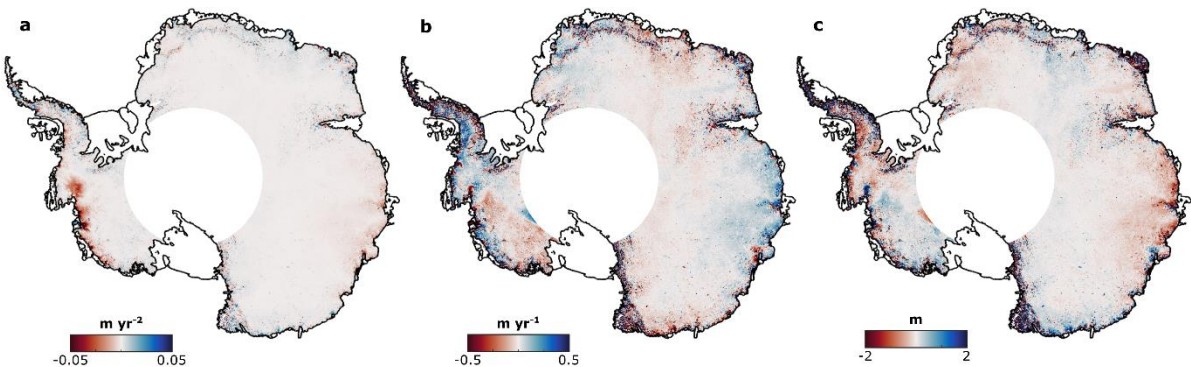

**Figure A1. Coefficients from a time-dependent quadratic polynomial fitted to surface elevation change observations.** The (a) acceleration, (b) linear and (c) intercept coefficients of the quadratic fit to observed surface elevation changes from 1992 to 2023 as measured by a suite of radar altimetry satellite missions.

This approach forces our 1996 to 2021 mean discharge derived from FrankenBedAdj to match the 1996 to 2021 mean discharge inferred from ice sheet surface elevation change measurements and regional climate model output. As such, the FrankenBedAdj discharge dataset is not fully independent of one altimetry-derived mass change estimate; therefore, we recommend caution if using that subset of the dataset in inter-comparison exercises such as IMBIE. The FrankenBedAdj estimate does, however, provide an indicator of which regions of existing Antarctic bed topography datasets may be under- or over-estimating the bed elevation on average, especially in drainage basins where there is confidence in the velocity measurements and SMB products.

Given that our adjustment is a simple proportional shift of the FrankenBed profile across each of the MEaSUREs glacier basins, we do not intend FrankenBedAdj to be taken as a superior bed elevation product for Antarctica. This proportional shift in ice thickness leads to greater cross-flow gradients in bed slope where thickness increases and lower gradients where

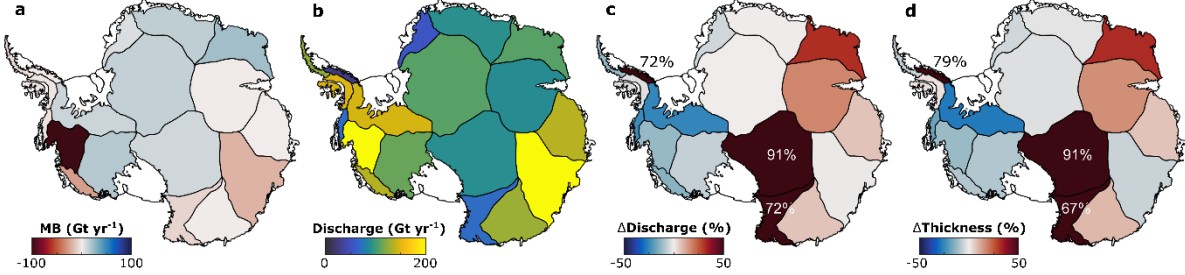

**Figure A2. Creating FrankenBedAdj.** (a) Observed rate of mass change from 1996 to 2021 based on the observed rates of ice surface elevation change in Figure A1. (b) Grounding line discharge, within each MEaSUREs regional basin, implied by the observed rates of mass change, given the surface mass balance from the mean of three regional climate models (described in text). (c) The percentage difference between the discharge in panel (b) and our average 1996-2021 discharge derived from FrankenBed. (d) The change in FrankenBed thickness required at the most downstream flux gate in order to reproduce panel (b).

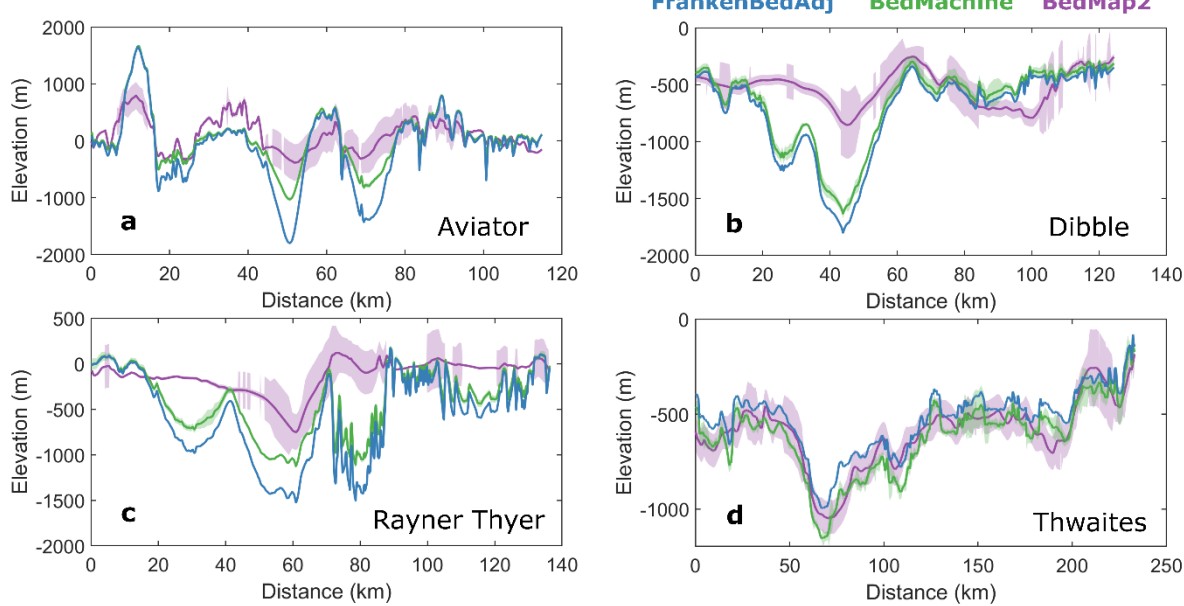

**Figure A3. Bed elevation profiles.** Each panel shows bed elevation in BedMachine v3 (green), BedMap2 (magenta) and FrankenBedAdj at our most seaward flux gate (blue) in example basins.

thickness decreases (Fig. 1; Fig. A3). Our approach will be somewhat sensitive to the choice of basins used to perform the integration and it produces unrealistic steps in bed elevation at the boundaries between basins. These steps have little impact on our grounding line discharge estimate, but could be consequential if the modified bed data were used in, for example, ice

sheet modelling applications.

## Appendix B: Sentinel-1 ice velocity maps

We generate monthly velocity mosaics from October 2014 through to January 2024 by applying standard intensity tracking techniques (Strozzi et al., 2002) to Copernicus Sentinel-1 synthetic aperture radar (SAR) single look complex (SLC) interferometric wide mode (IW) image pairs (Hogg et al., 2017; Davison et al., 2023b). We process all available 6- and 12-day image pairs acquired over Antarctica; all image pairs prior to the launch of Sentinel-1b in April 2016 and after the failure of Sentinel-1b in December 2021 are 12-day pairs. We estimate ice motion by performing a normalized cross-correlation between image patches with dimensions 256 pixels in range and 64 pixels in azimuth, and a step size of 64 pixels in range and 16 pixels in azimuth. To maximise tracking results in regions where velocity varies by more than an order of magnitude, we also use patch sizes of 362x144 and 400x160 pixels over East and West Antarctica, and four further patch sizes on the Antarctic Peninsula (192x48, 224x56, 288x72 and 320x80 pixels in range and azimuth). For scenes in East and West Antarctica, we use the a 1 km DEM (Bamber et al., 2009), whereas for scenes in the Antarctic Peninsula we use the REMA 200 m DEM (Howat et al., 2019). Prior to image cross-correlation, we perform image geocoding using the precise orbit ephemeris (accurate to 5 cm) where available and the restituted orbits otherwise (accurate to 10 cm) (Fernández et al., 2015). In common with comparable estimates of Greenland Ice Sheet velocity (Solgaard et al., 2021), we find no significant difference between pairs processed using each orbit type. Each image pair velocity field is posted on a 100x100 m grid in Antarctic Polar Stereographic coordinates (EPSG 3031).

For each image pair, we generate a signal-to-noise ratio-weighted mean velocity field of all available cross-correlation window sizes after removing outliers in the 2-D velocity fields. To remove outliers in each window size for every scene pair, we first compare each speed field to a reference speed map (Rignot et al., 2017); speed estimates more than four times greater or four times smaller than the reference map are considered outliers and removed. Secondly, flow directions more than 45 degrees different from the reference map are considered outliers and removed. Thirdly, pixels in which the speed differs by more than three standard deviations from its neighbours in a 5x5 moving window are removed. Similarly, pixels in which the flow direction differs by more than 45 degrees from its neighbours in a 5x5 moving window are removed. Finally, we use a hybrid median filter with a 3x3 moving window, which removes the central pixel if it more than three times the median of the horizontally and diagonally connected pixels. After forming the signal-to-noise-ratio (SNR) weighted mean of the resulting velocity fields, we generate Antarctic-wide mosaics of ice velocity for every unique date-pair since October 2014. From these date-pair mosaics, we generate monthly Antarctic wide velocity mosaics as the mean of all date-pairs that overlap with the target month. When doing so, we weight each date-pair by the number of days of overlap with the target month – in this way, 12-day pairs are weighted twice as much as 6-day pairs, which is appropriate because they should contribute more to the average velocity in the month. We also generate two quality parameters, the number of observations in each month in each pixel (after outlier removal) and the proportion of each month that is observed in each pixel, in addition to an error estimate defined the speed divided by the SNR (Lemos et al., 2018).

**Author contribution**

BJD designed the study, generated the Sentinel-1 velocity data, designed and implemented the discharge algorithm, wrote the manuscript and prepared all the figures. AEH acquired the funding and supported the Sentinel-1 velocity derivation. TS contributed the ice surface elevation change observations. RR provided technical support on all aspects of the Sentinel-1 velocity derivation. NH contributed to the discussion on mass balance and mass balance uncertainties. All authors commented on the paper.

**Competing interests**

The authors declare that they have no competing interests.

**Acknowledgements**

This work was undertaken on ARC4 and ARC3, part of the High Performance Computing facilities at the University of Leeds, UK. BJD gratefully acknowledges the European Space Agency and European Commission for providing Copernicus Sentinel-1 data, as well as all the research teams who generated the ice thickness, ice velocity, surface mass balance, firn air content and surface elevation change data that was used in this study, without which this research would not be possible.

**Financial support**

BJD and AEH are funded by: ESA via the ESA Polar+ Ice Shelves project (ESA-IPL-POE-EF-cb-LE-2019-834) and the SO-ICE project (ESA AO/1-10461/20/I-NB); NERC via the DeCAdeS project (NE/T012757/1); and by the UK EO Climate Information Service (NE/X019071/1). NH is funded by the Danish State through the National Centre for Climate Research (NCKF) and by the Novo Nordisk Foundation project PRECISE (NNF23OC0081251).

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
