# Peer review of "Antarctic Ice Sheet grounding line discharge from 1996 through 2023"

_Earth System Science Data, 2023_

## Referee Comment (RC1)

**REVIEW OF "Antarctic ice sheet grounding line discharge from 1996 to 2023"** *by Davidson et al., 2023*

Summary

This manuscript presents a 1996-present record of Antarctic-wide ice discharge at monthly resolution and finds that grounding line discharge increased by about 10% between 1996 and September 2023 (2205 Gt), mostly due to increasing flow speeds in West Antarctica. Several velocity and DEM products, at varying spatial and temporal coverage, are combined and iteratively gap-filled to create full reference maps. DEMs are differenced with multiple single (BedMap2, BedMachine v2) and hybridized ("FrankenBed") bathymetry datasets to compare the range of ice thickness estimates and net impact on ice discharge volume. In addition, a fourth reference bed, termed "FrankenBedAdj" represents the adjusted topography necessary to yield discharge volume changes compatible with altimetry-derived mass losses, when correcting for SMB. Grounding line discharge is calculated for a large variety of input variables, including bathymetry, as described above, multiple (2) firn compaction models, and flux gate location (16 equally spaced gate options plus the gate-average, for a total of 136 variations of ice discharge per drainage basin.

Overall, this is an interesting and both well written and executed study. The methods presented are robust and thorough – and the authors account for a variety of inputs and uncertainties to yield a comprehensive overview of grounding line discharge and its sensitivity to modeled + observational data and methods used. The manuscript will be valuable to the community and is thus deserving of publication in ESSD with minor revisions. However, I have also included suggestions for justification on several approaches, and a request to strongly consider a first order comparison to GRACE. Given the range in net discharge values discussed under various conditions in the manuscript, and regional discrepancies between these and previously published estimates, a comparison to gravimetry would provide valuable context for assessing sources of remaining error and uncertainty. My substantial comments are listed first, followed by minor comments and suggestions.

**Main:**

I would encourage the authors to provide useful context by comparing large basin mass balance estimates to GRACE. Given that a multi-model SMB average has already been incorporated into the study, combining these data with discharge for a mass balance time series for comparison to GRACE seems feasible and important. Offsets between GRACE/Input-Output time series, and their differences with respect to mass loss magnitude, long term trends, and seasonality would be valuable for better understanding sources of uncertainty and uncertainty in bed topography.

**Section 2.5: Thickness change between flux gates and grounding line.**
Please add more detail on this section. For example, on line 180:
"*For each gate pixel where ice flow is greater than 100 m/yr we calculate the number of years of ice flow between each flux gate pixel and the MEaSUREs grounding line (Mouginot et al., 2017c), to convert this rate of thickness change to a total thickness change.*"

Is the climatological SMB mean taken at the initial flux gate location, or an integrated average between the gate and MEaSUREs grounding line? Similarly, how is time of ice flow between gate and grounding line calculated? Lastly, is this correction applied to the mean gate, or each of the 16 gates?

I also struggle to understand the choice to compute the SMB-corrected ice thickness to better represent grounding line flux, but not also use the velocities at the grounding line. I may have misinterpreted the text, but the flux gates are placed three years of flow distance away from the grounding line, iterated every 0.1 years to account for variable velocity along the migration path. This implies that the velocity is different at the grounding line than at the most seaward flux gate. Why not use this velocity along with the corrected grounding line thickness to calculate grounding line discharge?

**Figure 10.**
This figure is really interesting, and Line 328 of the manuscript notes that "*The differences between flux gates primarily reflects the difficulty in conserving mass with imperfect ice thickness, velocity, and surface mass balance data, rather than algorithmic errors.*"

Do the values shown in Figure 10 reflect the location effect on discharge before or after the thickness adjustments (from surface processes, etc.) are applied? If the former, do these thickness adjustments account for the range in values seen across gates?

One recommendation is to apply the same methods discussed in Section 2.5 to the most inland gate, and compare to observed elevations at the downstream gate at the calculated between-gate flow time later. The difference between observed and calculated ice thickness change would provide uncertainty for the ice thickness adjustments discussed in Section 3.4.

Lastly, can you provide some discussion on the source of the seasonal "dimming" of inter-gate variability at the Peninsula (Figure 10, panel d). White patches reflecting similar discharge values across all gate locations seem to coincide with seasonal acceleration in ice flow (from Figure 9). However, the source of these patches prior to the availability of seasonally resolved dynamics (such as in 2006 through 2014) is unclear.

**On FrankenBedAdj**
On deriving an adjusted bed topography by "*proportionally adjust(ing) the pixel-based ice thickness based on the difference between our calculated basin*"
First, if this is an iterative process, what is the precision or threshold value used to determine satisfactory agreement? Second, does "proportionally" imply that every pixel is adjusted by the same +/- %? In instances where ice streams rest on deep troughs, this type of correction would tend to increase the cross-flow gradient in bed slope. Does the type of cross-flow pattern in bed shape calculated from the FrankenBedAdj appear realistic, given available direct observations from ice penetrating radar? It would be useful to see how these adjustments (along with the other three bathymetry estimates) compare at several select sites where direct observations of the bed are available. If FrankenBedAdj performs poorly at these locations compared to the other models, that would suggest the error likely lies in SMB/firn models.

**Minor**

**Line 159**
"*For gate pixels with no data at any time and more than 300 m from neighbouring finite pixels (after outlier removal), we use our reference ice velocity estimate which has no gaps by definition*."
What percentage of flux gate pixels fall into this category?

**Figure 9.**
Consider using larger markers for FrankenBed for panel (b) to avoid other lines completely obscuring the time series. I am also confused by the caption text "Note that FrankenBed and BedMachine are identical for all displayed basins except West." However, basin "Dryg" also indicates BedMachine (green) deviation from FrankenBed. How, if the beds are identical, do we see some divergence in the black and green time series in East Antarctica, panel c?

---

## Author Comment (AC1)

**Response to reviewer comments for the article "Antarctic Ice Sheet grounding line discharge from 1996 through 2023"**

**Overview of changes**

We would like to thank the reviewers and editor for their time and effort in reviewing our manuscript.

In response to the comments, we have:

(1) Included an estimate of mass change using our discharge estimate and compared it to an independent time-series (IMBIE3; Otosaka et al., 2023).
(2) Compared our modified bed topography (FrankenBedAdj) to the existing bed products, to illustrate the pattern and magnitude of thickness changes required to reproduce observed rates of mass change.
(3) Modified our approach for accounting for SMB-induced mass changes between the gate and the grounding line.

In addition, we have made some additional improvements to the algorithm, detailed below, and updated the time-series through January 2024.

Below, we have reproduced the comments from both reviewers in black with our response in blue. We have also appended a copy of the manuscript with the proposed changes.

**Responses to comments from Reviewer #1**

**Reviewer 1:** This manuscript presents a 1996-present record of Antarctic-wide ice discharge at monthly resolution and finds that grounding line discharge increased by about 10% between 1996 and September 2023 (2205 Gt), mostly due to increasing flow speeds in West Antarctica. Several velocity and DEM products, at varying spatial and temporal coverage, are combined and iteratively gap-filled to create full reference maps. DEMs are differenced with multiple single (BedMap2, BedMachine v2) and hybridized ("FrankenBed") bathymetry datasets to compare the range of ice thickness estimates and net impact on ice discharge volume. In addition, a fourth reference bed, termed "FrankenBedAdj" represents the adjusted topography necessary to yield discharge volume changes compatible with altimetry-derived mass losses, when correcting for SMB. Grounding line discharge is calculated for a large variety of input variables, including bathymetry, as described above, multiple (2) firn compaction models, and flux gate location (16 equally spaced gate options plus the gate-average, for a total of 136 variations of ice discharge per drainage basin.

Overall, this is an interesting and both well written and executed study. The methods presented are robust and thorough – and the authors account for a variety of inputs and uncertainties to yield a comprehensive overview of grounding line discharge and its sensitivity to modeled + observational data and methods used. The manuscript will be valuable to the community and is thus deserving of publication in ESSD with minor revisions. However, I have also included suggestions for justification on several approaches, and a request to strongly consider a first order comparison to GRACE. Given the range in net discharge values discussed under various conditions in the manuscript, and regional discrepancies between these and previously published estimates, a comparison to gravimetry would provide valuable context for assessing sources of remaining error and uncertainty. My substantial comments are listed first, followed by minor comments and suggestions.

We would like to thank the reviewer for their detailed and constructive comments on the manuscript. It is refreshing to receive a review that clearly shows a thorough and careful reading of the submitted manuscript, where the language is collegial and the comments constructive. We have endeavoured to respond in kind to each of the reviewer's major and minor comments below.

**Reviewer 1:** I would encourage the authors to provide useful context by comparing large basin mass balance estimates to GRACE. Given that a multi-model SMB average has already been incorporated into the study, combining these data with discharge for a mass balance time series for comparison to GRACE seems feasible and important. Offsets between GRACE/Input-Output time series, and their differences with respect to mass loss magnitude, long term trends, and seasonality would be valuable for better understanding sources of uncertainty and uncertainty in bed topography.

On submitting the manuscript, it was our intention to produce a follow-up paper and dataset providing mass balance estimates using this discharge dataset along with comparisons to independent estimates of mass change. In the revised manuscript, we have included a time-series of mass balance using our inverse error-weighted mean discharge for each bed topography dataset for Antarctica, West Antarctica, East Antarctica and the Peninsula. We have included a new figure (Figure 14) summarising those mass change time-series, including a comparison to the latest IMBIE dataset (Otosaka et al., 2023). We have also revised our discussion section "implications for mass balance" accordingly, focusing more on the differences in implied mass change between bed products in each region, and how each mass budget approach is expected to perform in each region. We think that goes some way to (1) providing another means to evaluate the quality our discharge dataset; and (2) allows some initial speculation about the sources of uncertainty in mass change estimates, as discussed in the revised text. We maintain however, that a thorough evaluation of the performance of each mass budget approach and improvements to measurements of mass change, will require closer examination of the uncertainties in each approach at smaller spatial scales than considered here. We think that

doing so would go far beyond what is reasonable to include in this paper and would require a full manuscript to achieve.

Otosaka, I. N., Shepherd, A., Ivins, E. R., Schlegel, N.-J., Amory, C., van den Broeke, M. R., Horwath, M., Joughin, I., King, M. D., Krinner, G., Nowicki, S., Payne, A. J., Rignot, E., Scambos, T., Simon, K. M., Smith, B. E., Sørensen, L. S., Velicogna, I., Whitehouse, P. L., A, G., Agosta, C., Ahlstrøm, A. P., Blazquez, A., Colgan, W., Engdahl, M. E., Fettweis, X., Forsberg, R., Gallée, H., Gardner, A., Gilbert, L., Gourmelen, N., Groh, A., Gunter, B. C., Harig, C., Helm, V., Khan, S. A., Kittel, C., Konrad, H., Langen, P. L., Lecavalier, B. S., Liang, C.-C., Loomis, B. D., McMillan, M., Melini, D., Mernild, S. H., Mottram, R., Mouginot, J., Nilsson, J., Noël, B., Pattle, M. E., Peltier, W. R., Pie, N., Roca, M., Sasgen, I., Save, H. V., Seo, K.-W., Scheuchl, B., Schrama, E. J. O., Schröder, L., Simonsen, S. B., Slater, T., Spada, G., Sutterley, T. C., Vishwakarma, B. D., van Wessem, J. M., Wiese, D., van der Wal, W., and Wouters, B.: Mass balance of the Greenland and Antarctic ice sheets from 1992 to 2020, Earth Syst. Sci. Data, 15, 1597–1616, https://doi.org/10.5194/essd-15-1597-2023, 2023.

**Reviewer 1: Section 2.5: Thickness change between flux gates and grounding line.**
Please add more detail on this section. For example, on line 180:

"For each gate pixel where ice flow is greater than 100 m/yr we calculate the number of years of ice flow between each flux gate pixel and the MEaSUREs grounding line (Mouginot et al., 2017c), to convert this rate of thickness change to a total thickness change."

Is the climatological SMB mean taken at the initial flux gate location, or an integrated average between the gate and MEaSUREs grounding line? Similarly, how is time of ice flow between gate and grounding line calculated? Lastly, is this correction applied to the mean gate, or each of the 16 gates?

I also struggle to understand the choice to compute the SMB-corrected ice thickness to better represent grounding line flux, but not also use the velocities at the grounding line. I may have misinterpreted the text, but the flux gates are placed three years of flow distance away from the grounding line, iterated every 0.1 years to account for variable velocity along the migration path. This implies that the velocity is different at the grounding line than at the most seaward flux gate. Why not use this velocity along with the corrected grounding line thickness to calculate grounding line discharge?

The last paragraph of the reviewer's comment made us question our choice here. On reflection, this was not the correct approach because, as the reviewer recognised, it produced something resembling the thickness at the grounding line but still used velocity from upstream. We have modified this approach so that we retain the thickness and velocity from upstream (after accounting for local changes in firn air content and ice surface elevation), but remove SMB-induced mass changes that occur between the gate and the grounding line. We do this by integrating RACMO2.3p2 SMB in the area between the gate and the grounding line for each basin. Since SMB is generally positive, this approach generally increases grounding line discharge. For Antarctica, it increases discharge by 64 Gt/yr on average compared to an approach that does not account for any SMB-induced mass change in that region. This approach also reduces the inter-gate differences between grounding line discharge to less than 1% for Antarctica as a whole (though this increases to at most 4.5 % at the highest elevation gates the Peninsula where the regional climate models struggle with the steep topography).

**Reviewer 1: Figure 10.** This figure is really interesting, and Line 328 of the manuscript notes that "The differences between flux gates primarily reflects the difficulty in conserving mass with imperfect ice thickness, velocity, and surface mass balance data, rather than algorithmic errors."

Do the values shown in Figure 10 reflect the location effect on discharge before or after the thickness adjustments (from surface processes, etc.) are applied? If the former, do these thickness adjustments account for the range in values seen across gates?

One recommendation is to apply the same methods discussed in Section 2.5 to the most inland gate, and compare to observed elevations at the downstream gate at the calculated between-gate flow time later. The difference between observed and calculated ice thickness change would provide uncertainty for the ice thickness adjustments discussed in Section 3.4.

Lastly, can you provide some discussion on the source of the seasonal "dimming" of inter-gate variability at the Peninsula (Figure 10, panel d). White patches reflecting similar discharge values across all gate locations seem to coincide with seasonal acceleration in ice flow (from Figure 9). However, the source of these patches prior to the availability of seasonally resolved dynamics (such as in 2006 through 2014) is unclear.

That description was quite brief and, somewhat embarrassingly, there clearly was an algorithmic error (or at least a poor choice of algorithm), which we have modified in the revised version. With this new approach to accounting for SMB, I don't think it would be informative to compare the thickness between gates – or at least we wouldn't expect them to match with this new algorithm.

The seasonal dimming in Figure 10 is an interesting observation. We looked at the individual components of inter-gate differences due to (1) changes in surface elevation, (2) changes in firn air content and, (3) changes in ice velocity. We have included a simple figure summarising that for the Antarctic Peninsula along one of the gates with large seasonality in the inter-gate difference (the most upstream gate). This shows (as expected) that changes in ice speed is the source of inter-gate differences in discharge. The integrate differences are generally greater in austral winter than in austral summer, within a hydrological year. We think this could reflect the greater reliance on interpolated, and therefore more stable velocities, in summer.

Regarding the apparent seasonality to the inter-gate differences in discharge prior to the availability of seasonal velocity data. This stemmed from the ~6-month offset between the annual measurements from ITS_LIVE and MEaSUREs, which differ slightly. In the revised manuscript, we have improved the alignment of velocity estimates, which reduces this apparent seasonality to the inter-gate differences in discharge.

[Figure]

**Reviewer 1: On FrankenBedAdj.** On deriving an adjusted bed topography by "proportionally adjust(ing) the pixel-based ice thickness based on the difference between our calculated basin"

First, if this is an iterative process, what is the precision or threshold value used to determine satisfactory agreement? Second, does "proportionally" imply that every pixel is adjusted by the same +/- %? In instances where ice streams rest on deep troughs, this type of correction would tend to

increase the cross-flow gradient in bed slope. Does the type of cross-flow pattern in bed shape calculated from the FrankenBedAdj appear realistic, given available direct observations from ice penetrating radar? It would be useful to see how these adjustments (along with the other three bathymetry estimates) compare at several select sites where direct observations of the bed are available. If FrankenBedAdj performs poorly at these locations compared to the other models, that would suggest the error likely lies in SMB/firn models.

Thanks for pointing out the typo here. This is indeed an iterative process and we used a threshold of 0.1 % of the discharge implied from SMB and mass balance. Note that we now perform this adjustment on the scale of the MEaSUREs regional basins, rather than the MEaSUREs glacier basins, because some of the latter are small compared to the resolution of the elevation change and SMB datasets.

Regarding the proportional modification of each flux gate pixel: yes, each pixel within the basin is modified by the same percent as others within the basin and the reviewer is correct that in some places this increases the cross-flow gradient in bed slope. Equally, in places where the bed elevation needs to increase on average, it decreases the cross-flow gradient in bed slope.

In the revised manuscript, we have included some example profiles of the resulting bed topography, with comparison to BedMachine and BedMap2. Given that BedMachine passes through the observed bed topography and that FrankenBedAdj is not intended to reflect the true topography, we don't think it would be beneficial to compare it with radar flight line data. All it would show is that FrankenBedAdj is different from the observations where they were incorporated from BedMachine, and that FrankenBedAdj and BedMachine differ where there are no observations. It would be fruitful to compare BedMachine to new bed elevation observations that have been collected since it was created, but we think that is tangential to the purpose of this study and will likely be addressed in part by the release of the BedMap3 gridded dataset (which we hope to incorporate into this discharge dataset).

**Reviewer 1 – minor comments**

Line 159: "For gate pixels with no data at any time and more than 300 m from neighbouring finite pixels (after outlier removal), we use our reference ice velocity estimate which has no gaps by definition."

What percentage of flux gate pixels fall into this category?

Between 0.09 and 0.16 % of flux gate pixels in the latest update. The areas that typically require filling are in very slow-flowing parts of the Peninsula near the coast, that are either unresolved, masked out or have poor velocity retrieval due to contamination with sea-ice (for example), and in the southernmost parts of the Ross Ice Shelf, which aren't surveyed by Sentinel-1. I've included a figure below to highlight them (in red) for our most upstream flux gate.

[Figure]

Figure 9. Consider using larger markers for FrankenBed for panel (b) to avoid other lines completely obscuring the time series. I am also confused by the caption text "Note that FrankenBed and BedMachine are identical for all displayed basins except West." However, basin "Dryg" also indicates BedMachine (green) deviation from FrankenBed. How, if the beds are identical, do we see some divergence in the black and green time series in East Antarctica, panel c?

We have increased the size of the markers for FrankenBed in this figure as suggested. The note in the text is a mistake. Drygalski Glacier uses the Huss and Farinotti bed topography dataset in FrankenBed, so does differ from BedMachine here as the reviewer points out.

**Reviewer 2:** The authors use a combination of datasets to estimate the grounding line discharge of Antarctica from 1996 to 2023. This is an extraordinarily complex problem because of the heterogeneities of the datasets, uncertainties in ice thickness, elevation changes, correction for thickness, location of flux gates, etc. and how error propagate to that chain, which means in practice that a blind approach to the problem is likely to fail to identify errors and/or their cause. Identifying local uncertainties is especially relevant to focus future observations.

Overall the study is well written, the details of the methodology are properly explained and relatively well thought through, including correction for polar stereo distance, but it is not clear that the mixing various models of unknown performance levels necessary yields a more reliable estimate of errors since model performance is not included in the weighting of the different inputs. The authors also offer no analysis of the changes in ice flux from their approach compared to other methods, so it is unclear where the differences are and what is their cause. As a general guideline for science, when updates of prior results are presented, it is a valuable exercise to compare with prior work, identify the differences, their impacts, and their sources. In this regard, the paper is lacking details. While I am not asking the authors to fill that void (that's their decision in the end), this gap will make it challenging to incorporate it in broader assessments which generally try hard to understand those details in order to generate further advances.

I have a few major comments and even fewer minor comments:

We are grateful to the reviewer for taking the time to review the manuscript. We are gratified that the reviewer found the manuscript well-written and the details of the methodology well-explained. We would like to state clearly here that we found it difficult to parse some of the reviewers concerns about the manuscript or to determine what change they would have liked to see in order to address their concerns. The reviewer appears to be concerned about our propagation of errors and the use of multiple datasets (or use of datasets that are themselves comprised of measurements from multiple types of sensor), but these concerns often make vague references to 'models' and 'errors' with providing any specific explanation of which aspects were unsatisfactory or made no recommendation for improvement. In these cases, we have attempted to determine the specific concern of the reviewer and have described in more detail the error components and their contribution to grounding line discharge, which in all cases were very small. Some of the reviewer's comments appear to reflect a misunderstanding of our methodology, so we have endeavoured to clarify the wording of our manuscript in those places. The reviewer also appears to be simultaneously concerned that we have not provided any comparison with prior work and that our comparison with prior work is too short. We have devoted a sub-section of the discussion to compare with every available Antarctic-wide grounding line discharge dataset that we are aware of and which we could access the data for (two of these were not downloadable and had to be digitised), which is substantially more comparison than any of those datasets provided. We think that comparison is thorough – there are comparisons for every basin provided by those other datasets, and clear quantification and visualisation of the spatial and temporal patterns to differences between those datasets and ours. Without access to the discharge components of the other datasets, we do not think that further comparisons would be fruitful.

**Major comments**

**Reviewer 2:** The Frankbed is essentially BedMachine v2 (BedMachine v3.7 is available, would be good to update the study with it). Huss and Farinotti (2014) is a (relatively old) model-based thickness, with unknown skills, because there are very few quality thickness data in the Peninsula other than at the GL based on flotation and precise surface elevation, so the error of Frankbed compared to BMv2 is unknown in the study.

We thank the reviewer for pointing out the new version of BedMachine, which we were not aware of. We have updated the dataset so that it uses BedMachine v3. We disagree with the reviewer's comment that "Frank[en]Bed is essentially BedMachine" – they are very different over the Peninsula and that difference creates a substantial change in discharge from Antarctica.

We are not sure what change the reviewer is recommending in the second part of their comment. We agree that there are very few thickness measurements on the Antarctic Peninsula, which is why the uncertainty in ice thickness is large here and why we think it is helpful to include a discharge estimate using each of BedMap2, BedMachine v3 and the Huss and Farinotti (2014) bed – we don't know which is closest to the true bed because we don't know the true bed. The use of the Huss and Farinotti (2014) dataset is further justified and perhaps even preferred over the other datasets because (1) the error in BedMachine and BedMap2 over the Peninsula is extremely high because there are so few thickness measurements; (2) in many places, the BedMachine and BedMap2 bed elevations are clearly unrealistic because they implies ice thicknesses of 10-20 m where the ice is flowing several hundred metres per year; (3) the Huss and Farinotti (2014) grid is derived from sound physical principles and, because of the regional scale of the dataset, was able to take advantage of high-resolution surface elevation data to inform the inversion. As such, the H&F grid produces much more reasonable ice thickness than does BedMachine v3 or BedMap2 given the observed ice speeds. We acknowledge that no one bed elevation dataset is perfect and that, without more measurements of the bed elevation, we can only use the errors provided in each product.

**Reviewer 2:** BMv2 already takes FC into account on the ice shelves, why remove FC again? What errors are introduced this way and is there a way the authors could evaluate their correction (or validate)?

This study only includes grounded ice. We assume FC means firn air content(?). The BedMachine v3 'surface' layer has the firn removed, but we make it clear in our methods that we use the BedMachine v3 bed and the REMA ice surface, which, to our knowledge, does not have firn removed. We do not remove the firn layer twice: we state clearly in our methods that we remove the climatological firn air content then remove anomalies in firn air content, so as to remove time-varying firn air content from the ice thickness estimates.

Regarding the evaluation of the firn correction, we already include a section in the manuscript titled "Effect of thickness adjustments" along with two figures, which clearly describe the effect of removing firn air content on grounding line discharge at different basins and throughout the time-series and with different firn models. We do this by calculating grounding line discharge both with and without the firn correction, so yes, there is a way we can evaluate the correction for firn air content, which we assume the reviewer's comment is concerned with.

**Reviewer 2:** SMB models have varying levels of performance in Antarctica, mixing their estimates could provide an indication of the uncertainty in SMB but does not really replace evaluating the skill of any of these models. RACMO2.3 is notoriously superior to other methods like MAR; and the HIRHAM is a notch below in Antarctica; but even various versions of RACMO2.3 disagree, especially in East Antarctica; and these differences are not discussed. There is also a specific version of RACMO2.3 for the Peninsula, but not sure the authors are aware.

Our aim is not to evaluate the skill of those regional climate models and we do not think that the reviewer's statement that "RACMO2.3 is notoriously superior to other methods [models?] like MAR" – one might be able to argue that the polar-optimized models (RACMO and MAR) better reproduce the limited SMB observations than HIRHAM, though often the differences are small or at least smaller than the uncertainties arising from the different spatial scales of in-situ SMB observations and regional climate model grids (see Mottram et al., 2021)

This study describes a dataset of grounding line discharge, for which an SMB correction is required to determine mass changes between the flux gate and the grounding line. Since we do not know the true SMB of Antarctica at the spatial and temporal scale, and at all points in time, required for this correction, we have to make recourse to regional climate models, which, partly because of the lack of observations, differ in their estimates and differ from the 'true' SMB by an unknown amount. We therefore think it is a reasonable approach to combine the SMB estimates from three RCMs to determine this correction. This correction contributes 64 Gt/yr on average to our discharge estimate for Antarctica compared to an approach that does not account for any SMB-induced mass change in that region, so it is an important component of this discharge estimate. However, the difference among the SMB models amounts to less than 1 % of our Antarctic-wide discharge, so the choice of SMB model here has almost no bearing on our grounding line discharge estimate. We are aware of the 5.5 km RACMO data over the Antarctic Peninsula.

Mottram, R., Hansen, N., Kittel, C., van Wessem, J. M., Agosta, C., Amory, C., Boberg, F., van de Berg, W. J., Fettweis, X., Gossart, A., van Lipzig, N. P. M., van Meijgaard, E., Orr, A., Phillips, T., Webster, S., Simonsen, S. B., and Souverijns, N.: What is the surface mass balance of Antarctica? An intercomparison of regional climate model estimates, The Cryosphere, 15, 3751–3784, https://doi.org/10.5194/tc-15-3751-2021, 2021.

**Reviewer 2:** FrankenBedabj includes corrections for match elevation changes over 1996-2014, which is a large source of uncertainty since altimetry over that time period mixes radar and laser instruments onboard various platforms, with varying level of performance, and surely low skills in steep areas, e.g. Peninsula. A comparison of this topography with actual data would help establish the actual performance level of this Frankenstein version of BedMachine. I could not find any element in the study that would convince me why this bed topography would be better than BMv2. It might very well be, but this has to be discussed more quantitatively.

We thank the reviewer for making a constructive suggestion to compare FrankenBedAdj to 'actual' data, by which we assume they mean measurements of bed elevation from, for example, radar surveys. In response to this and a similar comment from another reviewer, we now include a comparison between FrankenBedAdj, BedMachine v3 and BedMap2 in Figure 1 and Appendix A. Please see our detailed response to the first reviewer's comment for more information about this comparison.

Yes, there are uncertainties in mass change estimates from altimetry, especially in areas of steep topography, high accumulation and variable surface melt conditions. There are also uncertainties in snow and ice density estimates required for mass change estimates from elevation change observations, and uncertainties in the SMB estimates from regional climate models required for this inversion, which feed through into FrankenBedAdj. We are not sure how the reviewer has determined that the switch from radar to laser instruments in particular is a very large uncertainty in FrankenBedAdj – is the reviewer suggesting that time-averaged estimates of mass change during the radar-instrumented period and laser-instrumented period are accurate individually, but not in combination? In any case, we make it very clear in the manuscript that FrankenBedAdj is not a new bed topography dataset and should not be viewed as 'better' to the other datasets, it is simply the thickness required to reproduce the observed mass change. Indeed, in some places, that thickness seems unrealistic, which implies there are errors in the SMB or in the mass change estimate (assuming one is confident in the ice velocity data).

**Reviewer 2:** Later on, the authors correct REMA with altimetry 1992-2023, which introduces the same type of uncertainty at the surface, but they also correct for anomalies in SMB, but thickness is already corrected (twice if not once), so the entire picture is confusing. Would be nice to clarify.

This is one of the occasions where we think that the reviewer has misunderstood our algorithm. We shall clarify it here and we have revised the manuscript to try and clarify this, in order to reduce the risk of other readers making the same mistake.

We begin with an ice surface from REMA timestamped to May 2015 from which we remove the 30-year mean firn air content. We then remove time-varying anomalies (relative to the 30-year mean) in firn air content, which has the same effect as removing the firn air content through time. We then account for changes in ice surface elevation over time. We then remove SMB-induced mass changes between the flux gate and the grounding line – this final stage has changed in the revised manuscript and we have tried to clarify our description of the firn and surface elevation corrections, to avoid further confusion. The time-varying thickness changes have a very small impact on grounding line discharge (see Figure 11), so the reviewer's concern about the switch from radar to laser altimetry, which in itself is just one component of the error in the dH/dt time-series, will be a very small component of the discharge uncertainty.

**Reviewer 2:** The balance discharge from 3 x SMB models is problematic, especially in East Antarctica where RACMO2.3p2 has a lower performance than RACMO2.3p1, and in the Peninsula where only RACMO3.2 at 5.5 km has skills at reconstructing SMB. I do not have a sense of how these errors propagate in various estimates.

Please see our comment above about the various SMB estimates and, in general about uncertainties in mass change, ice thickness and SMB, and regarding the purpose of FrankenBedAdj, which is the only dataset affected by the balance discharge.

**Reviewer 2 minor comments**

Some comments seem arbitrary: "who manually selected different approaches to determine grounding line discharge and who modified ice thickness in a non-algorithmic manner in order to produce the required ice flux." Some specifics should be defined here rather than these blanket statements; not sure what "non-algorithmic manner" means, nor what is meant by "manual" approaches. This is a case where one would question: how do you know that your estimate is more precise than a prior one, and give us as many examples as possible for the most important basin. For instance, discussing uncertainties for Conger ice shelf is almost irrelevant. But discussing uncertainties for the Amundsen Sea Embayment sector of West Antarctica would seem to be a key priority.

We have modified this statement accordingly. We don't claim that FrankenBedAdj is more precise or better than any existing thickness dataset, it is simply the thickness required to reproduce the mass change derived from altimetry measurements, given the SMB from three RCMs. We highlighted basins and sectors in this discussion where there were large differences between discharge estimates and where they arose for different reasons – they are illustrative in order to provide the reader with an understanding of the differences between discharge estimates. In addition, Figures 9, 10 and 12 show the sensitivity of grounding line discharge in West Antarctica to the choice of bed topography dataset, gate location and firn model, which provide a lot of information to evaluate the impact of our choices on the dataset.

Line 125: It is unclear how the authors mix these velocity data from multiple sources into a cohesive dataset. How do the errors from these different products propagate? Monthly velocity maps at 100 m from velocity tracking seem almost impossible to achieve with S1. Why are they using ITS annual velocity maps but not the MEASURES annual velocity maps? Why is the multiyear map from 2020-2022 not used? Etc. The product quality varies significantly, so it is not clear what the aggregation of all of these estimates does to the analysis. An assessment should be made.:

1) 'Mixing' of velocity estimates from multiple sources: we clearly state that we assign each velocity measurement to a single time-stamp, align the different sources based on the mean difference of robust linear fits through each data source during their overlapping time-periods or, if the temporal resolution of each source is very different (annual or multiyear vs monthly, for example), then the average difference of the measurements themselves after temporal averaging of the monthly data.

2) How do the errors of each velocity product propagate? Again, we clearly state that we take the error from the respective datasets as provided. Where we have filled measurement gaps, we assume the error is 10% of the filled thickness. Since we never merge velocity measurements, there is no error propagation between measurements. We do propagate the individual velocity errors into our discharge error.

3) Monthly velocity maps at 100 m from Sentinel-1: the reviewer has not provided any argument or evidence to support this statement. The resolution of Sentinel-1 is nominally 2.3x14.1 m and we use a tracking window step size of approximately 100 m.

4) Use of ITSLIVE annual and MEaSUREs annual mosaics: this statement is incorrect – we do use annual velocity maps from MEaSUREs.

5) 2020-2022 MEaSUREs multi-year mosaic: we chose not to use this multi-year velocity mosaic because by that point we have monthly measurements with good coverage. Including a multi-year average velocity with lower resolution would create a velocity epoch that seemed much slower than the surrounding epochs, creating an apparent drop in discharge that would likely be erroneous.

6) The reviewer's final points are quite vague and the logic in the statement does not follow. As they did not provide any recommendation or suggestion of what such an assessment should look like or what even should be assessed, we are not sure how to respond to this. We already provide a value to indicate how the dataset alignment impacts Antarctic-wide grounding line discharge.

"Given that Antarctic Ice Sheet mass changes estimated from gravimetry and altimetry agree much more closely with each other than either do with the single available input-output mass change estimate (Otosaka et al., 2023)" ? This is not correct. Most of the disagreement is in East Antarctica. I would like to understand how the authors substantiate this comment.

This statement is based on Figure 3 of Otosaka et al. (2023). Yes, the differences between gravimetry and altimetry compared to input-output is greatest in East Antarctica.

Otosaka, I. N., Shepherd, A., Ivins, E. R., Schlegel, N.-J., Amory, C., van den Broeke, M. R., Horwath, M., Joughin, I., King, M. D., Krinner, G., Nowicki, S., Payne, A. J., Rignot, E., Scambos, T., Simon, K. M., Smith, B. E., Sørensen, L. S., Velicogna, I., Whitehouse, P. L., A, G., Agosta, C., Ahlstrøm, A. P., Blazquez, A., Colgan, W., Engdahl, M. E., Fettweis, X., Forsberg, R., Gallée, H., Gardner, A., Gilbert, L., Gourmelen, N., Groh, A., Gunter, B. C., Harig, C., Helm, V., Khan, S. A., Kittel, C., Konrad, H., Langen, P. L., Lecavalier, B. S., Liang, C.-C., Loomis, B. D., McMillan, M., Melini, D., Mernild, S. H., Mottram, R., Mouginot, J., Nilsson, J., Noël, B., Pattle, M. E., Peltier, W. R., Pie, N., Roca, M., Sasgen, I., Save, H. V., Seo, K.-W., Scheuchl, B., Schrama, E. J. O., Schröder, L., Simonsen, S. B., Slater, T., Spada, G., Sutterley, T. C., Vishwakarma, B. D., van Wessem, J. M., Wiese, D., van der Wal, W., and Wouters, B.: Mass balance of the Greenland and Antarctic ice sheets from 1992 to 2020, Earth Syst. Sci. Data, 15, 1597–1616, https://doi.org/10.5194/essd-15-1597-2023, 2023.

**Reviewer 2 conclusions:** Overall, the paper has value but glosses over the performance level of the various components used in the analysis to offer a holistic approach with little insights about differences, uncertainties, and causes. The comparison with prior estimates is too brief. Thickness ought to be a major one but the mixing of various velocity estimates could be another one, esp. given

the large uncertainty of ITS-Live coarse feature tracking. For major basins in West Antarctica and East Antarctica, the authors ought to compare their results with prior studies, identify the differences and explain them, but the results are very important to the ice sheet mass balance estimates. It would help justify why the current estimates are qualified by the authors to be "the best", which is amusing and unjustified.

We are disappointed that the reviewer thinks that we "glossed over the performance level of various components" of the dataset. We do not think that is a fair comment: we provide summary statistics, detailed description, discussion and figures to illustrate the impact on grounding line discharge of the choice of bed topography, firn air content, gate location and surface elevation change. We provide bulk estimates of the impact of velocity data source alignment on our grounding line discharge. We have provided an extensive comparison with several other grounding line discharge datasets. We also estimate the approximate changes in bed topography required to produce the observed mass changes, which provides important insights into the locations of uncertain bed topography, SMB and mass change, which are discussed in the manuscript (expanded on in the revised manuscript).

We are not sure what the reviewer means by ITSLIVE "coarse" feature tracking. As far as we are aware, ITS_LIVE use a similar window size to MEaSUREs and the ITS_LIVE mosaics are posted at 240x240 m, compared to 1x1 km for the MEaSUREs annual mosaics. So we are not sure what is meant by "coarse" feature tracking here and do not see a reason to exclude the ITS_LIVE data from our analysis.

The reviewer simultaneously claims that we have not compared our results with previous studies and that our comparison with previous studies is too brief. We provide a substantial figure and subsection devoted to comparing our dataset with other estimates. We have tried to be very transparent about our approach and the differences between our discharge estimates and those from other papers. We have provided illustrative examples of where the datasets differ and have explained why those differences occur.

By "the best" we meant that ours is the highest temporal resolution dataset and is the first to resolve changes in discharge on individual Antarctic Peninsula glaciers at monthly resolution, which we think is a real improvement over previous Antarctic-wide datasets (which often provided just one or two snapshots in time, or were at annual temporal resolution, and did not include as many basins, particularly on the Peninsula. We have modified the wording accordingly. We think our dataset offers potential for new discovery that would not be possible with those other datasets.

[revised manuscript text omitted]

---

## Author Response (AR2)

+ Abstract: "16 algorithmically-generated flux gates, which are continuous around Antarctica" Generally flux gates are per glacier or basin so this is confusing without reading the paper - which should not be required to understand the abstract. What about "16 algorithmically generated overlapping inter-comparable flux gates, which..." or something a bit longer to clarify that it isn't 16 outlets?

We have modified this sentence to: "We calculate ice flux through 16 algorithmically-generated flux gates across 998 ice sheet, glacier, ice stream and ice shelf drainage basins". We also note than in Antarctica, there is precedent for referring to flux gates in the manner we have here (Gardner et al., 2018).

Gardner, A.S., Moholdt, G., Scambos, T., Fahnstock, M., Ligtenberg, S., Van Den Broeke, M. and Nilsson, J., 2018. Increased West Antarctic and unchanged East Antarctic ice discharge over the last 7 years. The Cryosphere, 12(2), pp.521-547.

+ Zenodo URL: Maybe best to give the top/latest/permanent DOI but mention "this paper written using vX of the data" ?

Done

+ Does the pole hole (L112) cover any flux gates? If so, what % or weighted %? For the remainder of the velocity gap filling exercise, what % is filled? At L89 you mention the pole hole and 6 % weighted discharge contribution but this is for CryoSat-2, not velocity products.

Yes, this is a good point. The Sentinel-1 pole hole does cover some portion of the flux gates, amounting to 6.2 % of the Antarctic-wide discharge. We have added a note to that affect on line 167: "The forward-filling of the velocity time-series is used on all flux gate pixels south of 81.8° during the Sentinel-1 era, which contribute 6.2 % to our Antarctic-wide discharge". The ITS_LIVE and MEaSUREs velocity products are comprised of measurements from multiple satellites, so the pole hole for those aren't clear cut.

+ You describe "we use our reference velocity map" at line 166 near the end of 2.3, but this is 2+ pages after describing how you create it. Clarify why you generate a reference map earlier. Perhaps even why before how.

We agree that this was not an ideal structure. We have added a sentence at the beginning of section 2.3: "Prior to constructing an ice velocity and discharge time-series, we generate a reference velocity grid in order to fill gaps in the time-series products." Thanks for the suggestion.

+ L22: The Antarctic Ice Sheet is losing mass at an accelerating rate (Diener et al., 2021; Otosaka et al., 2023; Shepherd et al., 2019; Slater et al., 2021). But wasn't there mass gain last year?

Fair point. We have modified the wording of this statement to: "The rate of mass loss from the Antarctic Ice Sheet has accelerated since the early-1990s".

+ Fig 3: black dots show something? I don't see black dots. I see white dots, red dots, and blue dots.

Apologies, they are black circles, not black dots. We have modified the figure caption accordingly.

+ L152: Maybe (?) replace "typically indicate" with "have" (you started sentence with "generally" which may be a synonym with "typically" and therefore this could be repetitive).

Good spot. Done.

+ based on a robust linear fit <-- what does "robust" mean here?

'Robust' linear regression is less sensitive to outliers than standard linear regression (ordinary least-squares fitting). Robust regression uses iteratively reweighted least-squares to assign a weight to each data point, and it is less sensitive to large changes in small number of elements in the dataset. We chose to use a robust fit not an ordinary least-squares fit because we align the datasets before removing outliers, which we in turn chose to do to maximise the amount of time-series information available to identify, remove and replace outliers. We note on line 157 that the 'robust' fit is an iteratively re-weighted least-squares algorithm.

+ Outlier removal discussed 2x in separate paragraphs (L153, 158). Maybe join?. Split "gap filling" into new paragraph near L162. Are these gaps you create from outlier removal? Or other gaps due to assigning velocity products to time-stamps rather than time-spans?. Or both?

We have described this stage of the processing chronologically (remove gross outliers using the reference grid –> align all velocity sources -> then use full time-series information to remove remaining outliers -> then fill the gaps), which we think is clearer to the reader than the alternative, though of course one could make an argument alternative structures.

We have split the gap-filling into a new paragraph as suggested. The gap-filling is done on all gaps, whether or not they were originally present or if they were created by removing outliers. We don't think it's necessary to specify that here, given that we have already stated on line 150 that we create a gapless velocity time-series.

+ How do you calculate EPSG 3031 grid distortion? What is the mean/median/min/max?

We assume this refers to our section on flux gate pixel spacing, now on line 184. We use the methodology described in Numerical Weather and Climate Predication (Thomas Tomkins Warner, 2011, Cambridge University Press). m = (1 + sin(trueLat)) / (1 + sin(lat)), where m is the ratio of distance on a polar stereographic projection to distance on a sphere (where m<1, projected distances are shorter than those on a sphere), and trueLat is the reference latitude of the projection (-71 for EPSG 3031). We did not retain the un-corrected distances or the correction ratio for the flux gates, and we do not think it would be very helpful for the reader to provide summary statistics for the distortion given that we remove the effect of it.

+ Eq 6. Not sure I agree with the edits (changing variable names). The output of this function is

the discharge error (a function of velocity error, thickness error, and probably others not accounted for (see other comments), but inputs are velocity error and thickness error. What is "velocity-induced discharge error" isn't this just the "velocity error"? It is also the "velocity component of the discharge error", but no need to make it overly complicated.

1) We think it is worth distinguishing between error in the velocity and thickness measurements vs the resulting errors in the grounding line discharge because (for example) the velocity-induced discharge error resulting from a given velocity error is dependent on the ice thickness. The data-user is probably most interested in the velocity-induced discharge error and the thickness-induced discharge error, rather than the errors in the underlying thickness and velocity data.

2) We now use "velocity component of the discharge error" (also for thickness) as suggested.

We have substantially revised the section describing the derivation of the discharge errors to clarify our approach.

+ It also isn't clear to me where the "uniformly distributed random numbers" come from. That sentence explains "from time-stamped velocity error". But at each gate pixel at each time, how many velocity products (or velocity error products) exist? You list 8 at lines 127 -- 144 but not all exist everywhere or at all times, so I don't see how a normal distribution can be generated from them.

At each pixel and each time we have one measurement of cross-gate velocity error (incorporating the error in the vx and vy components of the velocity) and one measurement of thickness error (incorporating the bed error, surface error, firn error and dH/dt error). We want to know how those velocity and thickness errors impact grounding line discharge. One approach (used in your Greenland discharge datasets) is to add/subtract those errors from your central velocity and thickness measurement, to get upper and lower bounds on the grounding line discharge, given the errors in thickness and ice velocity. This approach therefore provides the discharge error under the assumption that the thickness and velocity value in every pixel could differ from the central estimate by the respective errors in the thickness and velocity. Our approach here is very similar, but we reasoned that the probability of every flux gate pixel being simultaneously different from the central estimate by the full error value was very small. Therefore, in every pixel and at all times, we take the velocity and thickness errors and say "those are the largest the errors can be in those pixels and at those times". Then, through 100 iterations, we randomly generate new thickness and velocity errors in every pixel and at all times, assuming those upper bounds from the original component errors. The random generation of errors assumes uniformly distributed random numbers, not normally distributed random numbers. This could potentially be improved if one had information about the form of the error in the underlying data, but that information is not always available. This gives us 100 discharge estimates in every pixel and at all times, which represents the spread in possible discharge measurements given the velocity and thickness errors. We actually end up with 2x100 discharge estimates (one only using the velocity errors and one using the thickness errors). We then take the standard deviation of those 2x100 measurements as the velocity and thickness components of the discharge errors. We then combine those components in quadrature to get the discharge error in each gate pixel and measurement epoch. The basin-integrated errors are just $((D_{max}-D) + (D-D_{min}))/2$ . As noted above, we have substantially revised the section describing the derivation of the discharge errors to clarify our approach.

+ Do you need to updated Eqs. 7 and 10 $V_{\sigma}$ and $H_{\sigma}$ to the new $D^2_{vel\_\sigma}$ and $D^2_{H\_\sigma}$ in Eq. 6 (although I don't agree with those variable names)?

These equations describe the underlying thickness and velocity error, not their impact on the discharge estimate, so we think those variable names are correct. We do not have an equation to describe the Monte-Carlo approach, but we think the text description is sufficiently clear.

+ Fig 1: Ice thickness increase of ~100 % between FB and FBAdj? This seems extreme and warrants discussion. Is "unrealistically thin" on L71 the only discussion of this? You add Fig. 1 panel E in response to reviewer comments. Mention Fig A3 here as it contains more samples.

We agree this is extreme. We do raise this in our section 4.2 "implications for mass budget estimates", where we write: "Given that the magnitude of the thickness adjustments in FrankenBedAdj exceed 50 % in some places, and that these are typically in locations where SMB and firn air content estimates from different regional climate models disagree the most (such as basin E-Ep in East Antarctica), we think that factors (1) and (2) likely contribute the majority of differences between mass budget approaches at the spatial scale considered here".

+ In your response to reviewers you repeatedly mention that FrankenBedAdj compensates for uncertainties in mass change (from altimetry), ice thickness, and SMB. What are the relative %? In the abstract you answer this with """The uncertainties in our discharge dataset primarily result from uncertain bed elevation and flux gate location, which account for much of difference between our results and previous studies.""" If it's not all ice thickness (assigned to bed elevation, as opposed to thinning), is *Bed* the correct name for this term? Or should it be assigned a more general 'scaling factor' term such as k or alpha? L341 mentions PIG has 1.5 % error due to gates, but SMB error is likely larger than that at PIG and everywhere (SMB models have errors >> 1.5 %).

Section 4.2 discusses this and related issues at length, but given that the uncertainties in SMB are unknown, we don't think it's possible to provide a definitive response to this and we don't think it would be meaningful to do this at a coarse scale because they are location specific (depending on the local accumulation rate, density of bed elevation observations, surface slope, surrounding topography, to name just a few). Indeed, if we knew the relative contributions with confidence everywhere, then we would not need FrankenBedAdj.

These points are somewhat separate from the sentence in the abstract quoted here, which focuses on the contributions to error in the discharge dataset and the magnitude of those errors relative to differences between our discharge estimate and other datasets – we have modified the abstract text to better reflect that.

The quoted figure for PIG is the relative difference in discharge between flux gates. We provided that to provide the reader with a means to evaluate the uncertainty in the grounding

line discharge associated with gate location, which in turn reflects the quality of the thickness, velocity and SMB data. It indicates the precision of the discharge estimate rather than the accuracy. We agree the SMB models likely have larger errors than 1.5 % - those errors are not well quantified and are certainly not available at every RCM pixel and time step – but that doesn't necessarily translate to a large spread in flux gate discharge estimates.

+ Fig 10: Seasonality appears pot-2016, presumably due to Sentinel and not a change in the system. Can you estimate annual discharge vs. discharge sampled only when pre-2016 observations occur (summer only?) and from this estimate if pre-2016 estimates are low or high, and/or adjust uncertainty?

We could certainly compare annually-averaged discharge estimates after with those before, but we do not think this would be very informative. The annual mosaics from MEaSUReS and ITS_LIVE are, as far as we can tell from the available documentation, derived from image pairs in both and winter (and some spanning full years, summer to summer, or winter to winter) using a range of sensors. As mentioned in the main text, there are also systematic differences between each of the velocity products, which may be due to aliasing seasonal signals, but could also be due to various choices in the product development.

Additional private note (visible to authors and reviewers only):
+ R2 raised some generally good concepts (e.g., "more detail, more error analysis"), but I understand why you found it challenging to reply. Still, I wonder if in revision you can better address any of their concerns. As one easy example, they question your comment

"""Given that Antarctic Ice Sheet mass changes estimated from gravimetry and altimetry agree much more closely with each other than either do with the single available input-output mass change estimate"""

To which you point to Otosaka 2023 Fig. 3 and agree that EAIS is the major source of disagreement. You could clarify that, and that WAIS has IO and Gravimetry in closer agreement that Grav v. Alt, rather than not addressing their comment in the revised text. There may be other places where you can address some of their comments, even if not all - and you did, for example removing 'best' comment near the end.

We have substantially revised this paragraph discussing the differences amongst mass change estimates in different locations, which we think helps to more completely address this comment from the reviewer.

---

## Author Response (AR3)

We would like to thank the three reviewers for reviewing our manuscript. One reviewer accepted the manuscript as is, one reviewer provided several constructive comments, the majority of which we have implemented in this revision, and one reviewer recommended rejection. In this revision, we have (1) included substantially more comparison with existing estimates of Antarctic ice sheet discharge; (2) renamed 'FrankenBed' to BM+HF14; (3) clarified a number of assumptions pointed out by reviewer 3; and (4) clarified the wording throughout the manuscript, especially where there was misunderstanding in the reviews. We have provided detailed responses to each of the reviewers' comments below.

**Reviewer 1**

| 1 | In this study, the authors force the grounding line
thickness to be such that the resulting discharge will
produce a mass loss similar to that recorded by
altimetry missions from ERS-1 to ICESAT (1994-2021),
radar and laser mixed, effectively forcing the mass
balance solution to agree with altimetry.                                                                                                                                                                            | This was one relatively small component of
our study, which we did to explore the
contributions to uncertainty in mass
balance.                                                                                                                                                                                                                                                                                                                                                                                                                                                                                                                                                                                                                                                                                                                                                                                                                                                                                                                                    |
|---|---------------------------------------------------------------------------------------------------------------------------------------------------------------------------------------------------------------------------------------------------------------------------------------------------------------------------------------------------------------------------------------------------------------------------------------------------------------------------------------------------------|-----------------------------------------------------------------------------------------------------------------------------------------------------------------------------------------------------------------------------------------------------------------------------------------------------------------------------------------------------------------------------------------------------------------------------------------------------------------------------------------------------------------------------------------------------------------------------------------------------------------------------------------------------------------------------------------------------------------------------------------------------------------------------------------------------------------------------------------------------------------------------------------------------------------------------------------------------------------------------------------------------------------------------------------------------------------------------|
| 2 | Given that the ice thickness has been measured
around a large share of Antarctica, especially for the
largest glaciers most significant contributors to sea
level rise, by Operation IceBridge and other airborne
ventures, it seems to difficult to justify why these
data should be migrated other than to force an
agreement with altimetry at all cost. Since the
migration is larger than the inherent uncertainty of
these data, the adjustment is impossible to justify. | This point conflates some independent
aspects of our processing. We migrated our
initial flux to create 16 regularly spaced flux
gates. This is similar to the approach that
Mouginot et al. (2014) applied in the ASE.
As far as we are aware, only one study
(Gardner et al., 2018) uses a flux gate that
is specifically placed to follow e.g.
Operation IceBridge flight lines. In contrast,
Rignot et al. (2019) appear to use the
grounding line as the flux gate and Mankoff
et al. (2020; Greenland) use flux gates some
distance upstream of the grounding line.
It's not clear that using OIB data directly
would be better than using gridded
products which all assimilate the OIB data.
To be clear, the migration is not to force
agreement with altimetry. As described
above, we do include an experiment in
which we query what the thickness would
need to be in order to reproduce the long-
term rate of mass change from altimetry,
given the observed velocity and modelled
SMB. |
| 3 | In addition, the authors employ 3 sets of SMB
models: 1) RACMO, 2) MAR and 3) HIRHAM5. These
models have various levels of accuracy and precision.
Given that RACMO2.3p2 and p1 have a published
record of being far superior to the others, it is not
clear why the mixing of three SMB models would
produce a better assessment of mass balance.                                                                                                                                    | Our study focuses on producing grounding
line discharge estimates, for which the SMB
(between each flux gate and the grounding
line) is a small contribution. The choice of
SMB model does very strongly affect the
resultant mass balance, which we include
as an additional form of validation and for
completeness. Contrary to the reviewer's
claim, it does not seem that there is
consensus in the RCM community as to
which of the various RCMs is superior (see
e.g. https://doi.org/10.5194/tc-15-3751-
2021) and they are of course in constant
development.                                                                                                                                                                                                                                                                                                                                                                                                                                                        |

**Reviewer 3**

| 1 | Renaming "FrankenBed" – I think the abbreviation
"BedMachine+HF14" would be a more appropriate way
to denote the bed product is strongly similar to
BedMachine (expect Antarctic Peninsula). At present, the
FrankenBed name obscures/rebrands the BedMachine
product.                                                                                                                                                                                                                                                                                                                                                                                                                                                                                                                                                             | Agreed and done.                                                                                                                                                                                                                                                                                                                                                                                                                                                                                                                                                                                                                                                                                                                                                                                        |
|---|---------------------------------------------------------------------------------------------------------------------------------------------------------------------------------------------------------------------------------------------------------------------------------------------------------------------------------------------------------------------------------------------------------------------------------------------------------------------------------------------------------------------------------------------------------------------------------------------------------------------------------------------------------------------------------------------------------------------------------------------------------------------------------------------------------------------------------------------------|---------------------------------------------------------------------------------------------------------------------------------------------------------------------------------------------------------------------------------------------------------------------------------------------------------------------------------------------------------------------------------------------------------------------------------------------------------------------------------------------------------------------------------------------------------------------------------------------------------------------------------------------------------------------------------------------------------------------------------------------------------------------------------------------------------|
| 2 | Removing BedMachine-only – I don't think the
BedMachine lines contribute much beyond the
BedMachine+HF14 lines in the plots. Therefore
BedMachine lines could be removed, as they are 90% the
same as BedMachine+HF14 line, and only vary for good
reason on the Peninsula.                                                                                                                                                                                                                                                                                                                                                                                                                                                                                                                                                        | We have removed BedMachine only from
all of the Antarctic-wide figures, but we
think the comparison on the Peninsula is
useful information. It illustrates the impact
of our choice to use HF14.                                                                                                                                                                                                                                                                                                                                                                                                                                                                                                                                                                                            |
| 3 | Vertical velocity profile – Equation 1 should contain some
assumption about the ratio of depth-averaged velocity to
surface velocity. For perfectly deformational ice flow, the
depth-average velocity is 0.8 of surface velocity. At
present, the implicit assumption is 1, or plug flow, which
means that basal sliding velocity is assumed to be equal
to surface velocity along the entire flux gate perimeter.
This is likely to be the case. See how
https://doi.org/10.1029/2001JD900033 estimate this
ratio on a gate by gate basis. The authors need to
explicitly say they assume plug flow at all their gates, or
make gate by gate assumptions, for example informed by
convex/concave surface elevation profiles indicative of
basal sliding
(https://doi.org/10.3189/172756505781829430). | We agree that this is an assumption in our
processing. We do already state this at the
end of Section 2.3: "As in previous studies
(Mankoff et al., 2019; Mouginot et al.,
2014; Mankoff et al., 2020), we assume the
depth-averaged velocity is the same as the
measured surface velocity." But we now
explicitly state this again in our description
of equation 1: "where V is the depth-
averaged gate-normal ice velocity
(assumed to be equal to the surface
velocity),"                                                                                                                                                                                                                                                                           |
| 4 | Removing "FrankenBedAdj" – I applaud the authors for
trying to also take the opportunity to improve existing
bed products, but I feel that their further adjustment to
BedMachine+HF14 would be an article in itself. In brief,
they seek to use mass continuity to solve ice thickness as
the residual of velocity, surface mass balance anomaly,
and transient ice thickness change. This is more complex
than the approach of BedMachine, which applies mass
continuity to just velocity and surface mass balance (not
SMB anomaly, and not transient dH/dt). It is promising,
but presently appears underdeveloped and documented,
especially in the absence of any description of how
vertical velocity profile impacts balance velocity in each
basin.                                               | We agree that creating a gridded bed
product in the style of "FrankenBedAdj"
(now BM+HF14 Adj ) would require more
documentation, perhaps in the form of a
separate publication. However, we only
seek to scale BM+HF14 across each flux
gate within each of the MEaSUREs regional
basins. This is a much smaller undertaking
than was BedMachine. It is not intended as
a replacement for or improvement over
BedMachine (as stated in Section 4.2 and
Appendix A). We have added further
clarification of the purpose of BM+HF14 Adj
in section 2.1 when it is first described: "We
emphasise that BM+HF14 Adj is calculated
only across the flux gates (rather than
gridded) and is not intended to be a new |

|   |                                                                                                                                                                                                                                                                                                                                                                                                                                                                                                                                                                                  | bed product. Instead, it merely provides an
indication of what the thickness would
need to be to reproduce observed rates of
mass change, given the observed ice
velocity and modelled surface mass balance
in each basin. Differences between
BM+HF14 and BM+HF14 Adj are therefore
indicative of bed elevation uncertainties,
SMB uncertainties, mass change
uncertainties and ice velocity
uncertainties". We think this is a useful
exercise that provides new insights into
sources of mass change uncertainty that
are not available only by using existing bed
products.                           |
|---|----------------------------------------------------------------------------------------------------------------------------------------------------------------------------------------------------------------------------------------------------------------------------------------------------------------------------------------------------------------------------------------------------------------------------------------------------------------------------------------------------------------------------------------------------------------------------------|----------------------------------------------------------------------------------------------------------------------------------------------------------------------------------------------------------------------------------------------------------------------------------------------------------------------------------------------------------------------------------------------------------------------------------------------------------------------------------------------------------------------------------------------------------------------------------------------------------------------------------------------------------------|
| 5 | Glacier density – I appreciate that 917 kg/m3 is the
theoretical density of ice, but this is clearly an upper limit
for ice crossing the grounding line. For example, at
Columbia Glacier, the depth-averaged bulk glacier density
downstream of ELA has been estimated to be as low at
750 kg/m3 (https://doi.org/10.1002/2015RG000504). I
wonder if the authors should explicitly say they assume
that bulk glacier density is not influenced by crevasses?
Or, alternatively, at least use a conservative bulk density
range like 900 +/ 15 kg/m3? | We agree that 917 kg m -3 is an upper limit
for ice crossing the grounding line.
However, to the best of our knowledge,
several other major grounding line
discharge studies for use an ice density of
917 kg m -3 (Rignot et al., 2019; Mankoff et
al., 2020; Mouginot et al., 2019). To
maintain comparability with these datasets,
we think it would be best to retain the
upper limit ice density. In the revised
manuscript, we have stated that the use of
917 kg m -3 is an upper bound that neglects
the effect of crevasses, and that discharge
would scale with ice density. |
| 6 | Temporal change statements – In multiple places, the
authors state difference between July 1996 and January
2024, or simply 1996 and January 2024. But they also
highlight an annual cycle in more recent data. It seems
wise to limit temporal change statements to the same
month, i.e. July-July or January-January, to avoid biasing
multi-annual changes with a seasonal aliasing.                                                                                                                                                                        | Agreed. We now ensure comparisons are
done as the reviewer suggests throughout
the manuscript (in places this entails taking
an annual average of the monthly discharge
estimates in the latter part of the
timeseries).                                                                                                                                                                                                                                                                                                                                                                                                                        |
| 7 | Rignot Comparison – In Figure 13i, I see many large basins
with differences of up to +/- 50%. It seems that the
apparent agreement at ice-sheet scale is underlain by
substantial spatially compensating differences. I think the
reader would benefit from a more thorough discussion of
these differences, at least identifying the main cause of
difference between the opposing West and East Antarctic
biases, and possibly also at the scale of larger glacier
catchments/systems.                                                                 | We agree that the differences between our
discharge estimates and those of Rignot et
al. (2019; henceforth R19) were
emphasised but not sufficiently explored.
We have made substantial changes to the
text of section 4.1 exploring these
differences and have added another figure
to illustrate the contributions to the
differences in each basin where they can be
calculated. Although there are differences
between our central discharge estimates
and that of R19, our respective estimates
generally often within error or disagree by                                                                           |

|   |                                                                                                                                                                                                                                                                                                                                                                                                                                                                                                  | a similar amount as do other studies. This
is true at the ice sheet scale as well as for
the majority of basins. There are 29 basins
in which our discharge estimates do not
agree within error with R19. R19 used SMB
to estimate discharge for 15 of those – for
13 of those 15, the spread in SMB
estimates between RACMO, HIRHAM and
MAR is greater than the difference between
our discharge and R19. For the remaining
14 where R19 actually used thickness and
velocity to calculate discharge, we present
the contributions from thickness and
velocity – velocity is generally the dominant
contributor and stems from (1) our use of
multiple velocity datasets; (2) different
filling approaches (linear in time here, vs
linear in space or nearest in time,
depending on the size of the gap, in R19),
and; (3) R19s use of a scaled reference flux.
The total difference between our estimates
and R19 in those 14 basins that used
thickness and velocity is 69 Gt/yr, the
majority of which (59 Gt/yr) stems from 6
basins – Whillans, MacAyeal, Foundation,
Evans, Crosson and Bindschadler ice
stream. For those basins, we show in
Appendix 3 that there is a large spread in
velocity estimates from different velocity
sources and that the coverage is low. |
|---|--------------------------------------------------------------------------------------------------------------------------------------------------------------------------------------------------------------------------------------------------------------------------------------------------------------------------------------------------------------------------------------------------------------------------------------------------------------------------------------------------|--------------------------------------------------------------------------------------------------------------------------------------------------------------------------------------------------------------------------------------------------------------------------------------------------------------------------------------------------------------------------------------------------------------------------------------------------------------------------------------------------------------------------------------------------------------------------------------------------------------------------------------------------------------------------------------------------------------------------------------------------------------------------------------------------------------------------------------------------------------------------------------------------------------------------------------------------------------------------------------------------------------------------------------------------------------------------------------------------------------------------------------------------------------------------------------------------------------------------------------------------------------------------------------------------------------------------------------------------------------------------|
| 8 | Partitioning – L40 states that the input-output method
can yield direct partitioning of mass changes between
SMB and discharge. This is technically incorrect. If you
only know the SMB and D today, you need to make an
assumption about the steady-state SMD and D, in order
to partition mass loss today into both those terms. Their
ideas around FrankenBedAdj show that the authors are
close to estimating a long-term steady-state discharge,
but not quite yet. | We fully agree with this statement and have modified our wording accordingly.                                                                                                                                                                                                                                                                                                                                                                                                                                                                                                                                                                                                                                                                                                                                                                                                                                                                                                                                                                                                                                                                                                                                                                                                                                                                                            |
| 9 | Changes through space/time – It would be help to see
total 1996-2023/24 change in velocity plotted with
change in thickness along the entire perimeter. Ideally, it
would nice to identify where along the perimeter the
competing effects of thinning and acceleration result in
net discharge decrease versus increase.                                                                                                                                                         | We agree that this could be an interesting
analysis, and it would be possible with the
datasets we hope to release. Indeed, we
provide that information already, though in
a less succinct form, in our summary
figures. However, we think that this type of
analysis goes beyond the type of dataset
description suitable for ESSD. We think it is
sufficient to summarise the effect of ice
thickness changes on grounding line
discharge as presented in Section 3.3 and
Figure 11.                                                                                                                                                                                                                                                                                                                                                                                                                                                                                                                                                                                                                                                                                                                                                                                                                                                  |

---

## Author Response (AR4)

Dear Editor,

Firstly, we would like to thank the editor for their patience whilst we revised the manuscript in response to the reviewer comments. We received a mixed set of reviews in this iteration. One reviewer (R2) was happy to accept the manuscript as is, one (R3) provided a set of clear and constructive revisions that we have largely implemented, and another (R1) recommended rejection. It has taken us a long time to respond partly due to personal circumstances of the lead author and because of the seriousness of the comments from R1.

One of the main concerns raised by the reviewers was the disagreement between our dataset and one existing dataset (Rignot et al., 2019; R19). Indeed, Reviewer 1 stated that "comparison with other published work [was] almost avoided in this paper" – this statement is evidently incorrect because we had an entire section and figure devoted to comparing our estimates primarily with R19. These differences remain in the revised manuscript. In response to the reviewer comments and your own concerns, we have expanded this section, which now accounts for measurement uncertainty, incorporates other discharge estimates, accounts for errors and attempts to quantify the source of the differences between our estimate and that of R19. We think that there is now more than enough detail for the reader to understand the differences between discharge estimates and to evaluate the utility of our new dataset. While it should not be mandatory for all papers to perform formal intercomparisons with all previously published work (IMBIE, ISMIP6, GlaMBIE, etc have shown this is a substantial program of work in its own right), we think it is worth noting that R19 did no such comparison with the few other estimates available at the time R19 was published.

It is critical to have discharge estimates at the drainage basin scale, and as such reconciling differences at this scale is also important. We have attempted to calculate the contributions to the differences between our discharge estimates and those of R19 at the basin scale. The contributions are mostly from different choices of ice thickness and bed topography data and from using different ice velocity sources. Some details remain hard to quantify without access to the code or underlying data from R19 – but there are notable differences between their gap filling routine and ours, which will contribute to the discharge differences. We have also included a new appendix to describe and quantify the differences in ice velocity between data sources, which will also contribute to differences in discharge between this study and R19. There are several basins where R19 did not use ice thickness or velocity to estimate discharge – instead they used surface mass balance from one regional climate model under the assumption that the steady-state discharge must balance the basin-integrated SMB in order to maintain mass balance. In these basins, we use three regional climate models to estimate what the uncertainty in that flux is – in most of those basins, this uncertainty is larger than the difference between our discharge and that of R19.

Both Reviewer 1 and Reviewer 3 felt that our choice of bed datasets was not what they would have used. Reviewer 3 felt that BedMachine did not add much info beyond BM+HF14 – this is a fair point in most cases, so we have removed it from figures where these datasets were identical. We opted to retain it as an output, because we think it is useful to show the impact of including the Huss and Farinotti (2014) dataset on the Peninsula. Reviewer 1 felt than BedMap-2 should not be used at all because it was not designed for mass conservation. This is a surprising comment because R19 use BedMap-2 in their discharge dataset. Arguably, BedMap2 is now used less by the community than BedMachine, so could potentially be excluded, but we think the comparison between the two may be interesting to some readers and some members of the community may benefit from the option of BedMap2, depending on their research question. Ultimately, we feel that some of the value in our paper is that we have openly and transparently set out what the impact of using different input datasets is on the discharge estimate.

Reviewer 1 raised a number of other concerns, mostly shared privately. Some of these concerns were clear and constructive, which we have taken on board and revised our manuscript accordingly or have attempted to justify our choice. For example, the reviewer disagreed with our choice to use gridded bed

elevation datasets extracted along arbitrary flux gates, rather than 'raw' Operation IceBridge data extracted along flight lines. Although it may be ideal to do the latter, we argued that the gridded datasets incorporate the 'raw' data sufficiently well, and account for distance to flight lines in their errors, that the ease-of-use of the gridded data outweighs the potentially small benefit conferred by using the flight line data directly.

Reviewer 1 was also strongly opposed to our use of a tuned bed topography dataset (previously FrankenBedAdj, now BM+HF14$_{Adj}$); specifically, they did not agree with the use of altimetry-derived mass change data to determine the tuning. Rates of mass change averaged over 30-years are similar between most altimetry and GRACE estimates, (they have been reconciled within errors in the latest IMBIE), except in mountainous areas like the Peninsula, so using another source of mass change data for the tuning would not greatly affect the tuning results. Reviewer 3 also felt that this tuning warranted more explanation, validation and discussion than was possible in this journal. Our preference is to retain this tuned dataset for the community to use, or not, as they see fit. In the revised manuscript, we have even more explicitly stated the assumptions and aims of the tuning, to avoid inadvertently misleading readers – it is not intended to represent a new bed product. It might prove to be a useful standalone discharge estimate for intercomparison purposes. As a minimum, it is a useful experiment to quantify how much the ice thickness would need to change in order to reproduce the observed rate of mass change. We argue that this is useful because it provides information that will help future researchers to reconcile mass balance estimates, when combined with information about SMB, firn compaction, elevation change and ice velocity uncertainties in each basin.

Reviewer 1 also disagreed with our use of multiple data sources. They are not clear exactly what data sets they are referring to. There are indeed large differences between each of the velocity datasets in some locations, and it would be worth rigorously evaluating, intercomparing and understanding these differences in future studies. In the revised manuscript we have improved our approach to aligning the velocity datasets, which has reduced our total grounding line discharge compared to the previous submission. We have also added a new appendix to illustrate the velocity differences and show our alignment approach. In the same vein, Reviewer 1 strongly disagreed with our use of multiple regional climate models (RCMs) everywhere, arguing that an arbitrary combination of RACMO2.3p1 and p2 (split spatially between some basins) is preferable. Firstly, for the discharge estimate, the RCMs are only used to estimate the gate-to-grounding line surface mass change. This is a relatively small correction that is not always used (Mankoff et al., 2019, 2020; Rignot et al., 2019) and is only very briefly described elsewhere (Mouginot et al., 2014). For mass balance calculations, the discussions presented in Mottram et al. (2021) show that the choice of RCM is more important. Mottram et al. (2021) demonstrated large differences between RCMs but no significant differences in performance compared to in-situ SMB measurements and therefore suggested that an average of all RCMs may be the best approach for approximating the true SMB. We think it is worth noting that the ice sheet total and basin scale SMB estimates have changed changed, sometimes substantially, every time there is a new version of each RCM, so it is unreasonable for any reviewer to insist that one particular version of a single RCM is used. We know from communications with the RCM community that Reviewer 1's preferred SMB model will soon be superseded by a new version (with ~200 Gt yr$^{-1}$ greater Antarctic SMB), which is in the process of being finalised and published.

Reviewer 1 argues that "information is plentiful", when referring to the number of grounding line discharge estimates for Antarctica. Reviewer 1 is the provider of the only discharge time-series for Antarctica, which is now seven years out of date. Only three other Antarctic-wide discharge estimates are available to our knowledge: one (Depoorter et al., 2013) provides a single number for the ice sheet, based on scaling the discharge from ice shelves; another (Gardner et al., 2018) provides two estimates (2008 and 2015) each for Antarctica, East Antarctica, West Antarctica and the Peninsula; the third (Miles et al., 2022) provides only discharge change and the data are not tabulated. Therefore, for one of the most important metrics of ice mass flow globally, the scientific community only has one excel

spreadsheet containing annual measurements up to 2017 and two tables in manuscripts containing a total of 9 measurements. Given that these measurements disagree everywhere except West Antarctica, there is clearly a need for more discharge datasets in the community and open discussion on how the measurements should be calculated. We think it is worth noting that the altimetry and gravimetry mass balance communities are substantially larger, with 7 and 15 mass balance estimates submitted to the community intercomparison exercise (IMBIE, 2018) from these techniques respectively. It is unfortunately necessary to question the motivation, and fair, impartial and objective judgement, of the lead author of the only available Antarctic discharge time-series when they appear to be seeking to limit the publication of any new discharge estimates.

Reviewer 1 generally seems to oppose our approach of providing multiple discharge estimates, stating that instead we should aim to provide the best possible estimate. This point is well taken and is certainly a desirable goal, but we do not think that there is sufficient validation data to determine what the best dataset is. There are lots of options, of which we have transparently explored many, but there is often not a convincing argument for using one combination of options as opposed to another. In the revised manuscript, we more clearly present BM+HF14 as our favoured bed/discharge dataset, but we don't see the value in removing the other datasets. The comparisons alone provide information that is not readily available elsewhere, and we feel the scientific community will value this information.

In summary, grounding line discharge is an important metric that can be used as part of investigations that can improve our understanding of recent drivers of ice flow variability and that can be used to estimate ice sheet mass change. Despite its importance, few estimates of grounding line discharge exist; those that do exist are now several years out of date and are, at best, annually resolved and do not resolve individual glaciers on the Peninsula. In addition, those estimates appear to disagree by an amount that can affect our conclusions regarding the direction of ice sheet mass change. In the revised manuscript, we have demonstrated that our discharge dataset performs at least as well as other datasets whilst providing additional benefits of being up-to-date and capturing more and smaller basins, especially on the Peninsula. It is provided in accessible formats and has clearly defined errors – admittedly, these should not be defining features, but that reflects the standard of existing Antarctic discharge datasets. There is clearly a desire in the scientific community for this dataset – our Zenodo repository has been downloaded over 1000 times – and we feel confident that our detailed revisions now mean that the dataset and manuscript are up to the high standards required by the community and ESSD.

Yours Sincerely,

Benjamin Davison

On behalf of the authors:

Benjamin J. Davison, Anna E. Hogg, Thomas Slater, Richard Rigby and Nicolaj Hansen

---

## Author Response (AR5)

**Response to reviews of " Antarctic Ice Sheet grounding line discharge from 1996 to 2024"**

We thank the reviewers and the editor for their time and effort in reviewing our paper, "Antarctic Ice Sheet grounding line discharge from 1996 to 2024", submitted for publication in ESSD. We welcome the positive feedback and insightful comments which we have endeavoured to fully address in this revision, and we hope you agree this improves the manuscript. We have incorporated the majority of the suggestions made by the reviewers (indicated by 'Done' at the start of our response). In the few cases where we have not implemented the reviewer's suggestion (indicated by 'comment' at the start of our response), we have provided a detailed description of the justification for each decision. The changes are highlighted in the manuscript through the track changes function. Please see below a point-by-point response to the reviewers' comments, where all line numbers refer to the revised manuscript file with the tracked changes.

1. Renaming FrankenBed – done in previous iteration

2. Removing BedMachine-only – I don't think the BedMachine lines contribute much beyond the BedMachine+HF14 lines in the plots. Therefore BedMachine lines could be removed, as they are 90% the same as BedMachine+HF14 line, and only vary for good reason on the Peninsula. [still in Figure 9].

Done. We have removed BedMachine only from the revised manuscript.

3. Vertical velocity profile – Equation 1 should contain some assumption about the ratio of depth-averaged velocity to surface velocity. For perfectly deformational ice flow, the depth-average velocity is 0.8 of surface velocity. At present, the implicit assumption is 1, or plug flow, which means that basal sliding velocity is assumed to be equal to surface velocity along the entire flux gate perimeter. This is likely to be the case. See how https://doi.org/10.1029/2001JD900033 estimate this ratio on a gate by gate basis. The authors need to explicitly say they assume plug flow at all their gates, or make gate by gate assumptions, for example informed by convex/concave surface elevation profiles indicative of basal sliding (https://doi.org/10.3189/172756505781829430). [the revised version just makes the assumption explicit but does not explore associated uncertainty (or propagate it)]

Done. Assuming surface velocities are equal to depth-averaged velocities to calculate ice flux is well established in the literature (Gardner et al., 2018; Rignot et al., 2019; Mankoff et al., 2020; Mouginot et al., 2014) and is supported by modelling studies e.g. "*fast sliding is not common just to fast-flowing features but is widespread on the continent*" (Morlighem et al. 2013).

In the previous version of the manuscript, we made our assumption regarding the use of surface ice velocity clear: "*As in previous studies (Mankoff et al., 2019; Mouginot et al., 2014; Mankoff et al., 2020), we assume the depth-averaged velocity is the same as the measured surface velocity*"; we thought this would address the reviewer's concern that "the authors need to explicitly say that they assume plug flow", which "is likely to be the case", but we apologise if this was not clear enough. In the revised manuscript, we have further clarified this assumption by stating that "*As in previous studies (Mankoff et al., 2019; Mouginot et al., 2014; Mankoff et al., 2020), **we assume plug flow** i.e. the depth-averaged velocity is the same as the measured surface velocity*".

4. Removing "FrankenBedAdj" – I applaud the authors for trying to also take the opportunity to improve existing bed products, but I feel that their further adjustment to BedMachine+HF14 would be an article in itself. In brief, they seek to use mass continuity to solve ice thickness as the residual of velocity, surface mass balance anomaly, and transient ice thickness change. This is more complex than the approach of BedMachine, which applies mass continuity to just velocity and surface mass balance (not SMB anomaly, and not transient dH/dt). It is promising, but presently appears underdeveloped and documented, especially in the absence of any description of how vertical velocity profile impacts balance velocity in each basin. [*No, this is declined. This is my biggest reservation, and likely a deal-breaker. I don't think maintaining the FrankenBedAdj residual without a comprehensive uncertainty assessment is a valid option.*]

Done. We acknowledge the reviewers' concern here and to address this we have removed "FrankenBedAdj" from the revised manuscript and dataset.

5. Glacier density – I appreciate that 917 kg/m3 is the theoretical density of ice, but this is clearly an upper limit for ice crossing the grounding line. For example, at Columbia Glacier, the depth-averaged bulk glacier density downstream of ELA has been estimated to be as low at 750 kg/m3 (https://doi.org/10.1002/2015RG000504). I wonder if the authors should explicitly say they assume that bulk glacier density is not influenced by crevasses? Or, alternatively, at least use a conservative bulk density range like 900 +/ 15 kg/m3? [*I see no response on this point, although the revised methods suggest it can be a 2% effect, it does not appear to be propagated*].

Comment. Using an ice density of 917 kg/m3 to calculate ice flux is well established in the Antarctic Ice Sheet literature (Gardner et al., 2018; Rignot et al., 2018; Mankoff et al., 2020). The density value of 750 kg/m3 suggested by the reviewer is taken from a study of a highly crevassed surging glacier, so is likely to be a outlier in terms of bulk density and is a very different flow unit to those in Antarctica. In addition to not following methodological precedent for density assumptions in the Antarctic science literature, if we were to use a different density value, then it would make the results presented in this paper far less directly comparable with other published work, so we have retained our 917 kg/m3 bulk density assumption.

In the previous version of the manuscript, we followed the reviewer's previous recommendation to explicitly state our assumption of bulk density: *"This is an upper bound on bulk ice density and does not account for the effect of crevasses lowering ice density near the grounding line. The effect of ice density on discharge is linear, so reducing ice density to, for example, 900 kg m-3 would reduce our grounding line discharge estimate by approximately 2 %."*.

Although it is easy to implement various bulk density values, there is not an obvious way to determine an appropriate value for ice density around the perimeter of Antarctica. The reviewer is correct that the presence of crevasses would lower the bulk density compared to an idealistic case of an ice slab entirely enclosing the crevasses. However, we use a 200x200 surface elevation product derived from 2 m elevation measurements, which does not necessarily provide the elevation along the top of the crevasses. We do not think, therefore, that it would be appropriate to attempt to determine an alternative bulk density value, because all such values would be little better than guesses. Overall, we think that clarifying our assumption and providing an example of how much the discharge estimate would change when using a more conservative bulk density value is a comprehensive response to the reviewer's comment, whilst also keeping our method consistent with the established literature.

6. Temporal change statements – In multiple places, the authors state difference between July 1996 and January 2024, or simply 1996 and January 2024. But they also highlight an annual cycle in more recent data. It seems wise to limit temporal change statements to the same month, i.e. July-July or January-January, to avoid biasing multi-annual changes with a seasonal aliasing. [they say these statements still remain, even at abstract level].

Done. All statements relating to changes in discharge now compare like-for-like – e.g. comparisons between 1996 and 2024 now use the annually-averaged discharge in 2024. In a few places, we describe the temporal coverage of our dataset (1996 to November 2024).

7. Rignot Comparison – [I feel this could still be improved with tabulated basin-scale comparison plus difference map. A XY scatter plot is not the most informative presentation. I don't see how the argument that R19 didn't have to do this is relevant.]

Done. In the revised manuscript, we have provided a tabulated basin-scale comparison between our dataset and other studies, over equivalent time periods. We already provide difference maps in Figure 14, which address this part of the reviewer's comment.

8. Partitioning – L40 states that the input-output method can yield direct partitioning of mass changes between SMB and discharge. This is technically incorrect. If you only know the SMB and D today, you need to make an

assumption about the steady-state SMD and D, in order to partition mass loss today into both those terms. Their ideas around FrankenBedAdj show that the authors are close to estimating a long-term steady-state discharge, but not quite yet.

Done. Our manuscript already explicitly states the assumption that the reviewer requests on line 37: "when combined with an estimate of steady-state SMB and discharge". We have removed FrankenBedAdj (BM+HF14$_{adj}$) from the revised manuscript, so the second part of the reviewer's comment relating to that is not relevant to this new version of the manscript.

9. Changes through space/time – It would be help to see total 1996-2023/24 change in velocity plotted with change in thickness along the entire perimeter. Ideally, it would nice to identify where along the perimeter the competing effects of thinning and acceleration result in net discharge decrease versus increase.

Comment. On this point we stand by our original response, that this type of scientific analysis is out of scope for ESSD because ESSD's Aims & Scope (https://www.earth-system-science-data.net/about/aims_and_scope.html) states "any interpretation of data is outside the scope of regular articles". We do, however, provide summaries of the contribution of thickness changes to discharge changes (Figure 11 for the whole ice sheet, and our supplementary figures provide this information for all Antarctic drainage basins).